
**Composition and mixing state of atmospheric aerosols determined by**
**electron microscopy: method development and application to aged Saharan**
**dust deposition in the Caribbean boundary layer**
Konrad Kandler[1], Kilian Schneiders[1], Martin Ebert[1], Markus Hartmann[1,+], Stephan Weinbruch[1],
Maria Prass[2], Christopher Pöhlker[2]
[1]Institute for Applied Geosciences, Technical University Darmstadt, 64287 Darmstadt, Germany
[2]Max Planck Institute for Chemistry, Multiphase Chemistry Department, 55128 Mainz, Germany
[+]now at: Experimental Aerosol and Cloud Microphysics Department, Tropos Leibniz-Institute für
Tropospheric Research (TROPOS), 04318 Leipzig, Germany
*Correspondence to:* K. Kandler (kandler@geo.tu-darmstadt.de)
**Abstract.** The microphysical properties, composition and mixing state of mineral dust, sea-salt and
secondary compounds were measured by active and passive aerosol sampling followed by electron
microscopy and X-ray fluorescence in the Caribbean marine boundary layer. Measurements were
carried out at Ragged Point, Barbados during June/July 2013 and August 2016. Techniques were
developed to conclude from collected aerosol on atmospheric concentrations and aerosol mixing-
state, and different models were compared. It became obvious that in the diameter range with the
highest dust deposition the models disagree by more than two orders of magnitude. Aerosol at
Ragged Point was dominated by dust, sea-salt and soluble sulfates in varying proportions.
Contribution of sea-salt was dependent on local wind speed. Sulfate concentrations were linked to
long-range transport from Africa / Europe and South America / Southern Atlantic Ocean. Dust
sources were in Western Africa. The total dust deposition observed was 10 mg m$^{-2}$ d$^{-1}$ (range 0.5–47
mg m$^{-2}$ d$^{-1}$), of which 0.67 mg m$^{-2}$ d$^{-1}$ was iron and 0.001 mg m$^{-2}$ d$^{-1}$ phosphorus. Iron deposition had
two sources, mainly silicate particles from Africa, and particularly in 2016 a lower contribution of
small iron-rich particles from South America or Barbados of probably anthropogenic origin. Dust
particles were mixed internally to a minor fraction (10 %), mostly with sea-salt and less frequently
with sulfate. It was estimated that average dust deposition velocity under ambient conditions is
increased by the internal mixture by 30–140 % for particles between 1 and 10 µm dust aerodynamic
diameter, with approximately 35 % at the mass median diameter of deposition (7.0 µm). For this size,
an effective deposition velocity of 6.4 mm/s (geometric standard deviation of 3.1 over all individual
particles) was observed.
**1  Introduction**
Mineral dust and sea-salt are globally the most abundant aerosol types in the atmosphere (Andreae
1995; Grini et al. 2005). They are considerably affecting the earth's radiation budged (Liao et al.
1998; Choobari et al. 2014) and have impact on cloud processes (Koehler et al. 2009; Tang et al.
2016; Karydis et al. 2017). Over the North Atlantic Ocean, large amounts of dust are transported
westwards in the Saharan Air Layer, until they reach the Caribbean (Karyampudi et al. 1999; Prospero
et al. 2014). Here, dust usually does not cross the Central American Dust Barrier (Nowottnick et al.
2011). Instead, it is down-mixed into the marine boundary layer by turbulent and convective
processes and removed from the atmosphere by wet and dry deposition processes. These processes
are not yet fully understood (Prospero et al. 2009; Nowottnick et al. 2011).



During its transport, mineral dust may undergo modifications by chemical processing, cloud
processing or microphysical effects (Andreae et al. 1986; Falkovich et al. 2001; Matsuki et al. 2005;
Sullivan et al. 2007a; Sullivan et al. 2007b). Different processes are expected to lead to different
modifications (e.g., Fitzgerald et al. 2015). These processes will change the composition and particle
size of dust, and thus modify its radiative properties and cloud impacts. To assess the mixing state of
mineral dust, techniques considering single particles are required. While there have been
investigations on dust mixing state in the past (Zhang et al. 1999; Zhang et al. 2004; Dall'Osto et al.
2010; Deboudt et al. 2010; Kandler et al. 2011a; Fitzgerald et al. 2015), the data basis is still limited.
In the present work, we present results from two field campaigns in summers 2013 and 2016, where
the aerosol in the marine boundary layer at Ragged Point in Barbados was collected by active and
passive sampling techniques.
A particular challenge for these campaigns was the high wind speed and the high humidities at the
sampling site. Therefore, the present publication consists of an extended methodical section with
three major topics and a methodical as well as atmosphere-related results section. One methodical
section deals with the determination of composition and mixing state of individual particles, taking
into account quantification artifacts and modeling the dust- and non-dust components as well as
their hygroscopic behavior. A second section is on particle collection representativeness and models
relating atmospheric concentration and deposition, taking into account the single particle properties
at ambient conditions. Finally, when aerosol mixing state is assessed based on offline aerosol
analysis, considerations on coincidental mixing have to be made to ensure the representativeness of
the results for the atmosphere. Therefore, in a third section these fundamental considerations based
on model as well as experimental data are presented. In the result section, we report first on these
theoretical and experimental methodical aspects, before we then discuss the atmospheric
implications of the measurements.

## 2 Methods

### 2.1 Particle sampling and location

Aerosol was sampled at Ragged Point, Barbados (13° 9' 54" N, 59° 25' 56" W) from June 14 until July
15, 2013, and from August 6 until August 28, 2016. Sampling was performed on top of the
measurement tower, approximately 17 m above the bluff (Zamora et al. 2011), which descends 30 m
to the sea surface. Particles were collected on pure carbon adhesive (Spectro Tabs, Plano GmbH,
Wetzlar, Germany) mounted on standard SEM aluminum stubs (Free-wing impactor, Dry particulate
deposition sampler) or pure nickel plates (cascade impactor).

### 2.1.1 Free-wing impactor (FWI)

A FWI was constructed for inlet-free collection of particles larger than 5 μm in diameter. A FWI
consists in general of a rotating arm with a sampling substrate attached, which acts as body impactor
(see Fig. S1 in electronic supplement). Rotation speed, wind speed and sample substrate geometry
determine the particle size cut-off for collection. FWI applied in previous investigations were
constructed with a rigid setup, so adaptation to actual meteorological conditions (i.e. perpendicular
adjustment of the impaction vector) needed to be performed by hand or was neglected (Jaenicke et
al. 1967; Noll 1970; Noll et al. 1985; Kandler et al. 2009). The present setup achieves perpendicularity
by self-adjustment of the flexibly mounted sampling substrate to the sum vector of wind and rotary
movement. This is performed by addition of a small wind vane at the rotating arm adjusting the angle



of the substrate. The rotating arm is driven by a stepper motor, which is mounted on a larger wind
vane, aligning the construction with the horizontal wind vector. To ensure that the wind vanes
respond only to the dynamic pressure, any imbalance in the setup must be avoided. The arm length
of the FWI is 0.25 m. With a constant rotation frequency of 10 Hz and the wind speeds at the
sampling location, particle impaction speeds between 16.4 and 20.2 m/s were achieved. This
corresponded to sampling volumes of air between 2.7 and 14.7 m³ for the present campaign. While
in principle the FWI could disturb its own flow field in low wind situations – the sample collector may
be influenced by its own wake from the previous rotation – this was not an issue for the present
work, as the distance of the sampling volume shifted by the wind between the same angular
positions of two consecutive rotations was always larger than 0.45 m. This is a large and therefore
safe distance in comparison to the small diameter of the sampling substrate and the counterweight
(12.5 mm and 25 mm, respectively). In total, 30 samples were collected during the campaign in 2013.

### 2.1.2   Dry particulate deposition sampler (DPDS)

The DPDS used in the present work is derived from the flat plate sampler of Ott et al. (2008b), which
performed best with respect to wind speed dependence in their tests. It consists of two round brass
plates (top plate diameter 203 mm, bottom plate 127 mm, thickness 1 mm each) mounted in a
distance of 16 mm. In contrast to the referred design, the one used here has a cylindrical dip in the
lower plate, which removes the sampling substrate – a SEM stub with a height of 3.2 mm – from the
airflow, reducing the flow disturbance. The dip is larger than the SEM stub and has small holes in the
bottom to catch and dispose droplets creeping across the lower plate due to the wind dynamical
pressure. The top surface of the SEM stub is located 5 mm below the lower plate's top surface.
Larger droplets (> 1 mm) are prevented by this setup from reaching the SEM stub surface at the local
wind speeds (Ott et al. 2008b). A total of 29 samples were collected in 2013 and 22 in 2016.

### 2.1.3   Cascade impactor (CI)

While the principle design of the used CI is described by Kandler et al. (2007), a new version with a
larger housing, but with the same collection characteristics , was deployed in the present work. An
omnidirectional inlet with a central flow deflector cone was used, whose transmission is discussed in
section 2.4.3. The impactor was operated at a flow rate of 0.48 l/min, which is set by a critical nozzle.
Nozzle diameters of 2.04, 1.31, 0.71, 0.49, 0.38, and 0.25 mm were used, corresponding to nominal
cut-off aerodynamic diameters of 5.2, 2.7, 1.0, 0.54, 0.33, and 0.1 µm, respectively. Sampling times
were adjusted to the estimated aerosol concentration and ranged between 10 and 60 min for the
supermicron and between 12 and 45 s for the submicron fraction. A total of 30 CI samples were
collected in 2013.

## 2.2   Meteorological data, backward trajectory analyses and high-volume sampling / mass concentrations

In 2013, meteorological data was obtained at Ragged Point directly next to the particle sampling
devices. In 2016 wind, temperature and relative humidity were measured in parallel at The Barbados
Cloud Observatory at Deebles Point, which is located 400 m across a small cove to the southeast
(Stevens et al. 2016).
The measurements in 2013 are grouped into two time periods divided by the passage of the tropical
storm Chantal, which changed the atmospheric structure and air mass origin (Weinzierl et al. 2017).
The period from June 14 to July 8 will be referred to as pre-storm, the one from July 10 to 15 as post-
storm.



Backward trajectories were calculated with Hysplit 4 rev. 761 (Stein et al. 2015) based on Global Data
Assimilation System (GDAS) with 0.5 ° grid resolution (NOAA-ARL 2017). A backward-trajectory
ensemble consisting of a grid of 3x3 trajectories ending at 13.16483 (± 0.5)° N and 59.43203 (± 0.5) °
W at each altitude above sea level (300, 500, 700, 1000, 1500, and 2500 m) was calculated. Backward
trajectory length was 10 days in 1-hour steps, and an ensemble calculation was started for every
hour during the sample collection periods. Taking into account particle concentrations and
deposition rates as well as chemical properties, potential source contribution functions (PSCF) were
calculated (Ashbaugh et al. 1985) with a boundary layer approach. For each trajectory point it was
checked, whether the trajectory altitude was below the lowest boundary layer height provided by
the GDAS data set. If this condition was met, this particular point was regarded as a potential aerosol
injection spot and counted into the according source grid cell of 1° x 1° size. For determining possible
sources, all trajectories originating during collection of a particular sample were attributed with
sample properties of interest. Finally, the average for each source grid cell was calculated and then
weighted with a function based on the number of points in the cell to avoid an overrepresentation of
cells with high statistical uncertainty. The weighting function is generalized from the step function of
Xu et al. (2010) as

$$wt_{PSCF} = \exp\left[-2.93\left(\frac{W_j}{\overline{W}} + 0.89\right)^{-2.94}\right]$$

(1)

with    $W_j$    the number of trajectory points counted in cell number $j$,

$\overline{W}$    the average number of trajectory points per cell.

As result, a map based on PSCF shows regions with typically high or low values for airmasses passing
through the boundary layer in according grid cells. Note that by this approach, sources contributing
to advected aerosol can be identified, but local sources of course will not provide a usable signal.
Also, aerosol from remote sources might be transported inside the boundary layer and, thus, would
be attributed to also to the transport path in addition to its source.
## 2.3    Scanning electron microscopy: individual particle composition,
##        analytical and statistical uncertainties
About 22,000 individual particles (FWI), 65,700 (DPDS) and 26,500 (CI) were analyzed with a scanning
electron microscope (SEM; FEI ESEM Quanta 200 FEG and 400 FEG, FEI, Eindhoven, The Netherlands)
combined with an energy-dispersive X-ray analysis (EDX; EDAX Phoenix, EDAX, Tilburg, The
Netherlands and Oxford X-Max 120, Oxford Instruments, Abingdon, United Kingdom). The samples
were analyzed under vacuum conditions (approximately $10^{-2}$ Pa) without any pretreatment. Before
automated analysis, samples were screened for surface defects, distinctive unusual particles shapes
or deposition patterns indicating possible artifacts or contamination, and traces of liquids. Areas with
surface defects (holes and bubbles in the substrate) were excluded from further data processing. The
remaining areas were free from artifacts. However, FWI samples suffered in general from the
presence of dried sea-spray droplets. Sample analysis was performed automatically by the software-
controlled electron microscope (software EDAX/AMETEK GENESIS 5.231 and Oxford Aztec 3.3).
Automated particle segmentation from the background was performed with the backscatter electron
signal. An acceleration voltage of 12.5 kV, a 'spot size 5' (beam diameter about 3 nm) and a working
distance of approximately 10 mm were used. Scanning resolution was adjusted to the particles size.
For the FWI and DPDS samples 140 to 300 nm/pixel were used, for the CI samples 180 or 360
nm/pixel for the stage containing the largest particles (mainly particles larger than 2.5 µm diameter)



and 73 nm/pixel for the stages containing smaller particles. An X-ray signal collection time between
15 s and 20 s (EDAX) and 2 s (Oxford) for each particle was used (yielding 40,000–100,000 total
counts), during which the beam was scanned over the particle cross section area.

The image analysis integrated into the SEM-EDX software determines the particles size as projected
area diameter:

$$d_g = \sqrt{4B/\pi} \tag{2}$$

with $B$ the area covered by the particle on the sample substrate.

Following Ott et al. (2008a), the volume-equivalent diameter is estimated from the projected area
diameter via the volumetric shape factor expressed by particle projected area and perimeter as

$$d_v = \frac{4\pi B}{P^2} d_g = \frac{1}{P^2}\sqrt{64\pi B^3} \tag{3}$$

with $P$ the perimeter of the particle.

In addition, for the assessment of particle coverage homogeneity and size distribution determination
series of 1,000 to 2,700 images were acquired for each sample. They were analyzed by the Software
Fiji/ImageJ 1.51d (Rasband 2015), using also Eq. (2) for particle size determination after application
of a "triangle" type auto threshold for particles segmentation (refer to Fiji/ImageJ documentation for
further details).

### 2.3.1 Quantification of elemental composition
Fully quantitative results in EDX analysis can only be reached under specific sample conditions. When
the composition of an analyzed spot is derived from an X-ray spectrum, the sample geometry has to
be considered. Besides assuming perfect smoothness and homogeneity, commonly either infinite
sample depth (i.e. significantly larger than the interaction volume of a few µm) or presence of an
infinitely thin film is assumed. In the former case, a 'ZAF' correction can be applied (Trincavelli et al.
2014), in the latter for example the Cliff-Lorimer method (Cliff et al. 1975). However, for particles
these assumptions and the resulting quantifications are not valid, as shown by Laskin et al. (2006) in
their Fig. 3. To overcome this problem, several standard-less techniques can be applied (Trincavelli et
al. 2014), for example a Monte Carlo simulation of the interaction volume can be used (Ro et al.
2003). Alternatively, particle ZAF algorithms can be applied at least for larger particles with diameters
above 1 µm (Armstrong 1991; Weinbruch et al. 1997). In the present work, however, an approach
with less computational cost is applied. First, on the measurement side, a lower acceleration
voltage–12.5 kV instead of 20 kV in comparative studies – eases the particle morphology problem.
Second, on the data analysis side by combining the above-mentioned correction methods as function
of particle size, a higher accuracy can be achieved. In principle, particle smaller than a limit size are
considered as thin films and particles large than a second limit are considered to be of infinite depth.
Between the limiting sizes, values are interpolated. To determine the best interpolation method,
samples with well-known composition (sodium chloride, albite) were milled to obtain particle
standards with sizes between 1 and 30 µm. Particles were dispersed in clean air, re-deposited on the
same sampling substrate and analyzed like described above. Several non-linear interpolation
schemes were tested; the best results were obtained with:





$$\langle X \rangle = \begin{cases} X_{CL} & d_g \leq d_l \\ X_{CL} + (X_{ZAF} - X_{CL}) \dfrac{\log(d_g/d_l)}{\log(d_u/d_l)} & d_l < d_g \leq d_u \\ X_{ZAF} & d_g > d_u \end{cases} \tag{4}$$

with   $\langle X \rangle$        the corrected concentration of a particular element in the
beam interaction volume,
$X_{CL}$        the element concentration determined by the Cliff-Lorrimer method,
$X_{ZAF}$       the element concentration determined by the ZAF method,
$d_l = 1.5\,\mu m$    the lower interpolation range size limit,
$d_u = 30\,\mu m$    the upper interpolation range size limit.

Note that the concentrations are always normalized to 100 % of the beam interaction volume, which
can include besides the particle also the substrate; i.e. they do only indirectly represent an amount of
matter with respect to the particle. The correction is identical for atomic and mass concentrations; in
the present manuscript, atomic concentrations are used unless otherwise specified.
The result of the correction as function of particle size is shown in Fig. 1. It becomes clearly visible
that the accuracy of the quantification is strongly improved, while the major uncertainty remaining
originates from the particle to particle variation. This uncertainty depends on the noise in the
analysis system, but is also related to particle surface morphology and its variability. The latter
affects the X-ray signal mainly by unknown absorption path lengths, particularly for the lighter
elements, as illustrated by Fletcher et al. (2011). The measurements shown here were performed at
20 kV acceleration voltage; at 12.5 kV as used for the sample analyses, the problems are considerably
less pronounced.
Application to a sample of atmospheric particles is shown in Fig. 2. Particles dominated by Na and Cl
were selected from all DPDS samples, and the positive and negative ion contributions were
calculated for each particle from the determined concentration. It becomes obvious, that for a wide
size range the applied correction works well and produces therefore unbiased relative
concentrations for the considered elements. The outliers may occur due to noise, the negligence of
C, N and O compounds or an internal mixture of sea-salt with dust (e.g., $NaAlSi_3O_8$, $FeS$).
Using the measured and corrected atomic concentrations, an element index is defined as:

$$|X_i| = \frac{\langle X_i \rangle}{\sum \langle X \rangle} \tag{5}$$

with   $|X_i|$    element index of a particular element with arbitrary index $i$,
$\sum \langle X \rangle$    sum of all considered elements (Na, Mg, Al, Si, P, S, Cl, K, Ca, Ti, Cr, Mn, Fe, Co,
if not stated differently).

**2.3.2   Analytical measurement errors**
A typical deposition sample (collected between June 21, 2013, 13:46 and June 22, 15:02, local time)
was analyzed 29 times with a signal collection time per particle of 16 s. The same 300 particles were
analyzed each time. For illustration of the typical precision, the particles consisting mainly of Na and
Cl were selected from. Fig. 3 illustrates the average composition and standard deviation (1 σ) for
each particle. The average values show a typical behavior for atmospheric sea-salt with a slightly
depleted Cl and enriched S concentration (e.g., McInnes et al. 1994). The precision – shown as
relative standard deviation – increases with particle size. This is caused by the increasing amount of



material contributing to the particle's signal up to a point at about 3 μm particles, from which the
beam excitation volume is completely inside the particle. For the major compounds, the precision is
in the range of 2 % relative standard deviation. For minor compounds, it is between 10 and 20 % for
particles 3 μm and larger, but can exceed 100 % for the smallest ones. The latter high uncertainties
could be decreased with suitable working conditions (magnification, measurement time), but are not
focus of the present paper.
The uncertainty in particle diameter also depends on its size. For particles with 2 μm diameter, the
relative standard deviation is about 1.5 % decreasing to less than 1 % for particles larger than 3 μm.
This is in the same range as the systematic accuracy of SEM (1–2 %).

### 251   2.3.3    Estimation of the dust contribution to each single particle in a dust / sea-salt / 252    sulfate mixture and the size of the according dust inclusion

Sampling was performed in a region where locally emitted sea-salt aerosol and other soluble species
are mixed with long-range transported mineral dust. As in particular the mineral dust contribution is
of special interest, dis-entangling the particle populations and considering them separately an
important task.
To calculate the size of a dust inclusion and the according volume fraction for an internally mixed
particle from the chemical composition, the different elemental contributions have to be attributed
to the dust or non-dust component. This analysis is restricted in the present work to the major
compounds only. For Al, Si, P, Ti and Fe it can be safely assumed that they belong to the dust
component, and S and Cl can be attributed to the non-dust component. Na, Mg, K and Ca, however,
are ambiguous and can be present in fractions. Therefore, a model is needed to estimate the
contribution of the ambiguous elements from the dust and non-dust component based on the single
particle chemical composition.
A problem arises here from the error in chemical quantification due to matrix composition and
particle geometry. While the correction outlined in section 2.3.1 adjusts the quantification accuracy
of the average particle composition, for single particles because of their unknown geometry and
surface orientation angles, still a considerable error in element quantification can occur. In particular,
a bias between light and heavier elements can be introduced by unaccounted X-ray absorption,
which can lead to under- as well as overestimation of the relative contribution of light elements
(Fletcher et al. 2011). As for the present aerosol the major cations ($Na^+$, $Mg^{2+}$) are light in comparison
to the major anions ($Cl^-$, S of $SO_4^{2-}$), a quantification bias will lead to an error in component
attribution. Particularly, an overestimation of the light elements will yield – by attribution of the ion
balance excess to the dust component – to an overestimation of the dust contribution. Therefore,
two model pathways are applied: an upper limit estimate, where a possibly overestimated fraction of
the ambiguous elements is attributed to the dust component, and a lower limit estimate, where all
ambiguous elements are attributed to the non-dust component.
The following assumptions are made:

1.   There is exactly one dust inclusion in each mixed particle

2.   Carbonaceous matter does not contribute

3.   Ca contributes to dust as carbonate

4.   Ca contributes to non-dust as sulfate / chloride

5.   Fe contributes to dust as $Fe^{3+}$

6.   S contributes as sulfate





7.  Na, Mg, Al, Si, P, K, Ti and Fe contribute to the dust according to their oxide weights
8.  N-containing compounds contribute only in case of a non-neutral ion balance as ammonium

and nitrate

9.  Dust density is $\rho_{dust} = 2650 \frac{kg}{m^3}$, non-dust density is $\rho_{nondust} = 2200 \frac{kg}{m^3}$ , averaged from

typical dust and non-dust constituents: illite, kaolinite, muscovite, quartz, albite, microcline,

calcite, gypsum, halite, sodium sulfate minerals in different hydratation states, and

mascagnite (Deer et al. 1992; Warneck et al. 2012)

***Estimation of the upper limit***
Following the above-listed assumptions, the apparent cation/anion charge ratio is defined as

$$r_{cat} = \frac{\sum cations_{charge}}{\sum anions_{charge}} \tag{6}$$

with   $\sum cations_{charge} = |Na| + 2|Mg| + |K| + 2|Ca|$, apparent sum of cation charges,
and   $\sum anions_{charge} = |Cl| + 2|S|$ , apparent sum of anion charges.
Note that $|X|$ denominates the concentration of element $\langle X \rangle$ given as atomic (i.e. molar) fraction
relative to the sum of all quantified element concentrations with the exclusion of O and lighter
elements.
If $r_{cat} > 1$, it is assumed in the upper limit estimate that the excess in the apparent sum of cation
charges is produced by the dust contribution. Thus, the dust contribution is calculated as the ion
balance excess as

$$c_{dust} = \frac{r_{cat} - 1}{r_{cat}} = \frac{\frac{\sum cations_{charge}}{\sum anions_{charge}} - 1}{\frac{\sum cations_{charge}}{\sum anions_{charge}}} = \frac{\sum cations_{charge} - \sum anions_{charge}}{\sum cations_{charge}} \tag{7}$$

Cation excess
If $c_{dust} > 0$, an equal fraction of each element's apparent cation contribution excess is attributed to
dust, i.e. the ion charge balance is virtually neutralized for the non-dust component. The dust and
non-dust masses are calculated as (see also Table S1 in electronic supplement)

$$m_{dust} = \sum dust_{oxides} + c_{dust} \sum cations_{oxide} \tag{8}$$

with   $\sum dust_{oxides} = Al_{oxide} + Si_{oxide} + P_{oxide} + Ti_{oxide} + Fe_{oxide}$,
and   $\sum cations_{oxide} = Na_{oxide} + Mg_{oxide} + K_{oxide} + Ca_{carbonate}$.
Note that stable sulfates (gypsum / anhydrite, alunite) are assigned to the non-dust component.

$$m_{nondust} = (1 - c_{dust}) \sum cations_{mass} + \sum anions_{mass} \tag{9}$$


with   $\sum cations_{mass} = Na_{mass} + Mg_{mass} + K_{mass} + Ca_{mass}$,
and   $\sum anions_{mass} = Cl_{mass} + SO_{4,mass}^{2-}$.
The mass contributions are calculated as shown in Table S1 in the electronic supplement.





Cation deficit
If $c_{dust} < 0$, i.e. there is a cation deficit, the missing cation is assumed to be ammonium. The dust
and non-dust masses are then calculated as

$$m_{dust} = \sum oxides \tag{10}$$

$$m_{nondust} = \sum cations_{mass} + \sum anions_{mass} + NH^+_{4,mass} \tag{11}$$

For calculation of the ammonium mass $NH^+_{4,mass}$ see Table S1 in the electronic supplement.
***Estimation of the lower limit***
The dust mass for lower limit estimate of the dust contribution is calculated according to Eq. (10).
The non-dust mass is calculated for $c_{dust} < 0$ according to Eq. (11). For $c_{dust} > 0$ nitrate is assumed
to be the missing anion and the non-dust mass is calculated as

$$m_{nondust} = \sum cations_{mass} + \sum anions_{mass} + NO^-_{3,mass} \tag{12}$$

Refer to Table S1 in the electronic supplement for calculation of the nitrate mass.
***Calculation of the dust fraction***
From the dust and non-dust mass contributions, the dust volume contribution to the particle is
calculated as

$$f_{dust} = \frac{\frac{m_{dust}}{\rho_{dust}}}{\frac{m_{dust}}{\rho_{dust}} + \frac{m_{nondust}}{\rho_{nondust}}} = \frac{m_{dust}}{m_{dust} + \frac{\rho_{dust}}{\rho_{nondust}} m_{nondust}} \tag{13}$$


and the diameter of the resulting dust inclusion as

$$\frac{\pi}{6} d^3_{v,dust} = f_{dust} \frac{\pi}{6} d^3_v \rightarrow \quad d_{v,dust} = f^{\frac{1}{3}}_{dust} d_v \tag{14}$$


The model outlined here by suffer from systematic errors:
1. In the presence of larger amount of nitrate and ammonium or organics, the dust contribution will
be overestimated, as the regarded composition is fitted to the apparent particle volume. However, in
Barbados the concentration of these compounds is usually small in comparison to the dust (Lepple et
al. 1976; Savoie et al. 1992; Eglinton et al. 2002; Prospero et al. 2009; Zamora et al. 2011).
2. The density values are averages for the assumed components, and the real density of a particle
may be smaller or larger. However, the density range for the components in question is small (dust:
2300 to 3000 kg/m³, non-dust 1800 to 2600 kg/m³ at maximum), so the error is considered to be less
than 10 %.
3. The mass contribution is estimated by ion charge balances. If for the ambiguous elements an
inhomogeneous distribution of univalent and bivalent elements exists (e.g., univalent like Na favoring
the non-dust component and bivalent like Ca favoring the dust component), an error of less than 5 %
in diameter can occur. With an assumption of 5 % iron content in dust, the maximum error due to
the $Fe^{3+}$ assumption is less than 0.2 % in diameter.





The upper and lower estimates yield diameters, which differ for the dust core diameter in average by
25 %; for 75 % of the particles the difference is less than a factor of two. From the analytical errors in
ratios for major compounds (less than 10 % systematically and 6 % repetition uncertainty), an dust
core size uncertainty of about 6 % is estimated, as long as the core is larger than 10 % of the particle.
An overall analytical uncertainty of 15 % relative core size is estimated. In conjunction with the
upper/lower limit estimates, an overall core size error of 25 % is considered appropriate.
*Estimation of a geometrical iron-availability index*
Iron bioavailability in general is depending on different chemical and microphysical parameters as
well as residence time in chemically aggressive environments (Shi et al. 2011a; Shi et al. 2012). If
considering a homogeneous iron distribution in larger and smaller particles, it seems plausible that
the distance to the surface – therefore the surface to volume ratio – should have an impact on the
short-term iron accessibility (e.g., Baker et al. 2006; Shi et al. 2011b). Therefore, as first order
estimate we define a geometrical surface iron availability index SIAI (after virtual dissolution of the
soluble compounds) as

$$SIAI = \frac{Fe_{oxide}}{m_{dust}} \, 4\pi d_{v,dust}^2 \qquad (15)$$

**2.3.4   Particle classification and relative ion balance**
For assessing the abundances and counting statistics of certain particle types, the particles were
classified into different groups and classes. Based on the element index and additional elemental
ratios, a set of rules used in former mineral dust investigations in a marine environment was applied
therefore. For details refer to Kandler et al. (2011a).
In addition, a relative ion balance is defined for single particles as:

$$IB_{rel} = \frac{\langle Na\rangle + 2\langle Mg\rangle + \langle K\rangle + 2\langle Ca\rangle - \langle Cl\rangle - 2\langle S\rangle}{\langle Na\rangle + 2\langle Mg\rangle + \langle K\rangle + 2\langle Ca\rangle + \langle Cl\rangle + 2\langle S\rangle} \qquad (16)$$

A positive relative ion balance – i.e. an excess of positive ions – would indicate an undetected
presence of negative ions like $NO_3^-$ or $CO_3^{2-}$, a negative one such of $H^+$ or $NH_3^+$, which all can't be
(reliably) quantified by EDX. The relative ion balance is calculated only for particles classified into the
soluble sulfate or sea-salt classes (see below for classification scheme).
**2.3.5   Statistical uncertainty of total volumes / masses and relative number abundances**
**from single particle measurements**
When assessing the uncertainty of values based on counted occurrences, frequently the counting
statistics are assumed to follow a Poisson distribution. However, when calculating total aerosol
masses or volumes, besides the measurement errors in particular the – usually few – large particles
can introduce a considerable statistical uncertainty, which is not necessarily accounted for by the
distribution assumption. Therefore, estimates of the statistical uncertainty based on single particle
counts for an a priori unknown frequency distribution (i. e. the counting frequency distribution
modified by the also unknown particle size distribution) either require reasonable assumptions or
distribution-independent estimators. In the present work, the uncertainty is estimated by a
bootstrap approach with Monte Carlo approximation (Efron 1979). Furthermore, the results of the
generally robust bootstrap approach (Efron 2003) are compared to a more simple approach, where
the counting statistics is assumed to follow a Poisson distribution. The Gaussian error propagation is
then calculated for the latter case.





For the bootstrap approach, a considerable number of data replications are necessary (Carpenter et
al. 2000; Pattengale et al. 2010). On the actual number, different recommendations exist with more
than 1000 being among the most common (Carpenter et al. 2000). As higher numbers lead to smaller
errors in the uncertainty estimate, 10,000 replications for each sample were performed in the
present work.
For the Poisson approach, with a counting error of $\Delta n = 1$ for a single particle count ($n = 1$) the
Gaussian error propagation of the standard deviation for a sum of particle volumes $V_k$ resolves to

$$\Delta V = \sqrt{\sum_k \left(\Delta n \frac{\partial}{\partial n} nV_k\right)^2 + \sum_k \left(\Delta V_k \frac{\partial}{\partial V_k} nV_k\right)^2} = \sqrt{\sum_k V_k^2 + \sum_k \Delta V_k^2} \tag{17}$$

with    $n$        the number of particles with Volume $V_k$, in this case always 1,
$\Delta V_k$    the volume measurement error,
$k$        the index for the single particles.

Similar considerations apply for the mass calculations.
The two-sided 95 % confidence interval is estimated for the Poisson distribution case as 1.96 times
the standard deviation, and for the bootstrap case as the 0.025 to 0.975 quantile range of the
bootstrap replications (bias corrected and accelerated method; DiCiccio et al. 1996; Carpenter et al.
2000).

Considering only the statistical uncertainty from Eq. (17), the distribution-based approach can be
compared to the bootstrap approach in terms of relative statistical uncertainty for the volume
estimated from two methods (Fig. 4). Clearly, the Poisson assumption underestimates the lower limit
of the two-sided 95 % confidence interval (i.e. overestimates the uncertainty), providing even
physically meaningless negative numbers. In contrast, the bootstrap approach yields most probably
more precise estimates (see also Efron 2003). For the upper limit of the interval, the Poisson
approach seems to underestimate the uncertainty, in particular with respect to the high volumes
which can be present in single particles (Fig. 4, left). When restricting the size range to particles of 1
µm to 20 µm in diameter (Fig. 4, right), as expected the differences in confidence interval limits
become much smaller and stay mostly below 20 % difference between the two approaches. Note in
particular the impact of the volume in the single largest particle. For the present work, the bootstrap
approach is preferred.
For the assessment of the confidence interval of relative counting abundances, frequently a
confidence interval based on a binomial distribution is used as estimate (Agresti et al. 1998), i.e. for a
relative number abundance of a certain particle type class $r$ the two-sided 95 % confidence interval is
approximated as (Hartung et al. 2005)

$$\mathrm{CL}_{0.025, 0.975} = \frac{3.84 + 2r \mp \sqrt{3.84\left(3.84 + 4r\frac{n-r}{n}\right)}}{2(n + 3.84)} \tag{18}$$

with    $r$    the count of particles in that class,
$n$    the total number of particles.
The two approaches show much closer agreement here than in the previous case (see Fig. S2 in
electronic supplement). Note that if the common Wald confidence interval is used (Agresti et al.





1998), with lower absolute particle numbers in a class, an increasing tendency of
over/underestimation similar to the previous case occurs up to meaningless negative values in the
binomial case. For sake of consistence, in the present work also for the relative abundances the
robust bootstrap approach for estimation of the confidence intervals was chosen.

### 2.4  Collection efficiency and deposition velocity relating atmospheric concentrations to deposition rates


#### 2.4.1  Determining the size distributions from the free-wing impactor measurements

Obtaining the atmospheric size distribution and representative contributions of particle populations
with different hygroscopicity from the FWI requires a set of corrections. These corrections are
applied to each single particle as a function of its size and composition and the thermodynamic
conditions during sampling by weighting its count with the product of all correction functions. First, a
window correction accounting for the exclusion of particles at the analysis image border is applied
(Kandler et al. 2009):

$$c_w = \frac{w_x w_y}{(w_x - d_p)(w_y - d_p)} \qquad (19)$$

Second, the collection efficiency of the FWI has to be regarded. Therefore, the ambient particle
diameter at the time of collection has to be estimated by accounting for the hygroscopic particle
growth:

$$d_{amb} = d_v g_{hyg} \qquad (20)$$

with $g_{hyg}$ the hygroscopic growth factor.
Hygroscopic growth can be estimated from the hygroscopicity parameter $\kappa$ (Petters et al. 2007) as

$$g_{hyg} = \left(1 + \frac{a_w}{1 - a_w} \kappa\right)^{\frac{1}{3}} \qquad (21)$$

with $a_w$ the water activity.
As only super-micron particles are considered in this part of the study, the water activity can be
equated with the relative humidity given as fraction. The hygroscopicity parameter can be
determined as volume-weighted average of the hygroscopicity parameters of the major contributing
components (Petters et al. 2007). Assuming a mixture of sodium sulfate and sodium chloride as the
components dominating the hygroscopic growth and assigning the dust component zero
hygroscopicity, the hygroscopicity parameter is approximated from the volume contributions as

$$\kappa = (1 - f_{dust}) \frac{0.68 \times Na_2SO_{4,volume} + 1.12 \times NaCl_{volume}}{Na_2SO_{4,volume} + NaCl_{volume}} \qquad (22)$$

For the calculation of the volume contributions, refer to Table S1 in the electronic supplement.
The collection efficiency $E(P)$ is parameterized (see below) from the experimentally determined
values for discs given by May et al. (1967) as a function of impaction parameter $P$:

$$P = \frac{S}{D} \qquad (23)$$





with     $S$     stopping distance,

$D$     characteristic dimension, here 12.5 mm.

While $P$ equals to the Stokes number within the Stokes regime, in the current work the particle
Reynolds numbers are considerably higher. In this regime, in analogy to Hinds (1999) the stopping
distance can be approximated with better than 3 % accuracy as

$$S = \frac{\rho_{amb} d_{amb}}{\rho_a \sqrt{\chi}} \left[ Re_p^{\frac{1}{3}} - \sqrt{6} \tan^{-1}\left( \frac{Re_p^{\frac{1}{3}}}{\sqrt{6}} \right) \right] \tag{24}$$

with     $\rho_{amb}$     ambient particle density, estimated from chemical composition and growth factor,

$\rho_a$     air density,

$\chi$     aerodynamic shape factor.

Results of the trigonometric function must be given as radian. The dry aerodynamic shape factor is
assumed as constant similar to Ott et al. (2008a), but is interpolated for particles mixed with water as
function of the hygroscopic growth factor:

$$\chi = \begin{cases} 1 + (\chi_0 - 1)\left( 1 - \frac{(g_{hyg} - 1)}{g_{hyg,lim}} \right) & g_{hyg} < g_{hyg,lim} \\ 1 & g_{hyg} > g_{hyg,lim} \end{cases} \tag{25}$$

with     $\chi_0 = 1.4$     an estimated dry shape factor (Ott et al. 2008a)

$g_{hyg,lim} = 1.3$     a hygroscopic growth factor at which the particles are assumed to be

spherical.

The particle Reynolds number is

$$Re_p = \frac{\rho_a v_i d_{amb}}{\eta \sqrt{\chi}} \tag{26}$$

with     $v_i = \sqrt{v_r^2 + v^2}$     the impaction velocity

$v_r = 2\pi l f_r$     the speed of the collector in the plane of rotation

$l$     the collector arm length

$f_r$     the rotation frequency

$v$     the wind speed

$\eta$     the viscosity of the air.

The stopping distances calculated by Eq. (24) are well in accordance with the parameterization
curves shown by May et al. (1967).
The collection $E(P)$ efficiency for $P > 0.125$ is then parameterized (see Fig. S3 in the electronic
supplement) and the according correction is

$$c_e = \frac{1}{E(P)} = \exp\left( \frac{0.28}{P} \right) \tag{27}$$

The total investigated volume for the concentration calculations is determined by

$$V_i = A \, v_i \, t_i \tag{28}$$





with   $A$      the analyzed area,
$t_i$    the sample collection time.

The atmospheric concentration is finally

$$C(d_{amb}) = \frac{1}{V_i} \sum_k c_w(d_{p,k}) c_e(d_{amb,k})$$

(29)

with   $k$      index of the particle.
Potential systematic error sources for this calculation are mainly the uncertainty in collection
efficiency, given the considerable spread in data points in the according literature (Golovin et al.
1962; May et al. 1967), and any bias in particle size.

**2.4.2   Determining the airborne size distributions from the sedimentation sampler**
**measurements**
Similar to the previous section, sampling efficiency considerations are necessary for the
sedimentation sampler. For the supermicron particle size range sedimentation and turbulent
impaction dominate the particle deposition velocity (as for example illustrated by Piskunov 2009). To
calculate the turbulent impaction velocity, which depends of the wind speed, the friction velocity is
needed. As the opposing inner boundary layers of the sampler plates are always separated for the
considered range of wind speeds (boundary layer thickness between 4.5 mm and 2 mm for wind
speeds between 3.5 m/s and 13.5 m/s; Munson et al. 2013), the flow inside the sampler is
approximated as flow over a smooth flat plate (the lower plate). The friction velocity is calculated as
recommended by Wood (1981):

$$u^* = \frac{v}{\sqrt{2}} (2 \log_{10} Re_s - 0.65)^{-1.15}$$

(30)

with   $Re_s = \frac{\rho_a v x}{\eta}$   the flow Reynolds number at the sampling stub location,
$x$      the distance from the lower plate edge to the center of the sampling stub
(6.3 cm).

Considering the flow inside the sampler as tube flow (Liu et al. 1974) would lead to friction velocities
differing by less than 5 %.
A variety of models estimating the particles deposition speed were published (Sehmel 1973; Slinn et
al. 1980; Noll et al. 2001; Wagner et al. 2001; Aluko et al. 2006; Piskunov 2009; Petroff et al. 2010).
They yield considerable different results, possibly due to negligence of unaccounted forces (e.g., Lai
et al. 2005), the way of determining the relevant friction velocity, or other model assumptions. For
the present work, the formalism of Piskunov (2009) was selected, as it derives the deposition velocity
rather from physical principles instead of parameterizing a specific measurement setup. The
deposition velocity is estimated by the following formalism:

$$v_d = \frac{u^*}{J_1 + J_2}$$

(31)

$$J_1 = \frac{u^* \exp(-1.2\tau^+)}{v_{Stk}} \left[ 1 - \exp\left( -13.204 \, Sc^{\frac{2}{3}} \frac{v_{Stk}}{u^*} \right) \right]$$

(32)



with $Sc = \frac{\eta}{\rho_a\, C_D} = \frac{3\pi\, \eta^2}{\rho_a\, k_B\, T} \frac{d_{amb}}{C_c\sqrt{\chi}}$  the Schmidt number,
$C_D$  the particle diffusion coefficient,
$k_B$  the Boltzmann constant,
$T$  the ambient temperature,
$C_c = 1 + 2\frac{\lambda\sqrt{\chi}}{d_{amb}}\left[1.257 + 0.4\exp\left(-\frac{1.1\, d_{amb}}{2\lambda\sqrt{\chi}}\right)\right]$  the Cunningham slip correction,
$\lambda = \frac{k_B T}{\sqrt{2}\,\pi\, d_M^2\, P}$  the mean free path,
$d_M = 3.68 \times 10^{-10}$ m  the average diameter of an air
molecule,
$P$  the ambient pressure,
$v_{Stk} = \frac{12\,\eta\sqrt{\chi}}{0.42\, C_c\, \rho_a\, d_{amb}}\left[\sqrt{1 + \frac{0.42\, C_c^2\, \rho_a\, \rho_{amb}}{108\, \eta^2}\left(\frac{d_{amb}}{\sqrt{\chi}}\right)^3\left(1 - \frac{\rho_a}{\rho_{amb}}\right)g} - 1\right]$
the gravitational settling velocity,
$g$  the gravitational acceleration;

$$J_2 = \frac{1 - \exp\left[-\gamma\left(1 + \frac{v_{Stk}}{u^* p_\tau}\right)\right]}{p_\tau + \frac{v_{Stk}}{u^*}} \qquad (33)$$

with $\gamma = \frac{0.4611\, Sc\, \tau^+(1+0.3859\,\tau^+)}{(1+0.1193\,\tau^+)(1+0.1193\,\tau^+ + 6.613\, Sc)}$
$\tau^+ = \frac{u^{*2}\rho_a}{\eta}\frac{v_{Stk}}{g}$  the dimensionless relaxation time,
$p_\tau = \frac{\tau^+(1+0.3859\,\tau^+)}{65.06\,(1+0.1193\,\tau^+)^2}$
The atmospheric concentrations are then

$$C(d_{amb}) = \frac{1}{A\, t_i}\sum_k \frac{c_w(d_{p,k})}{v_d(d_{amb,k})} \qquad (34)$$

A major bias for this calculation originates from the uncertainty in (turbulent) deposition velocity.
The deposition velocity calculated by different formalisms for a series of deposition samples is shown
in Fig. 5. The aerodynamic diameter used here is calculated as:

$$d_a = \sqrt{\frac{\rho_{amb}}{\rho_0 \chi}}\, d_{amb} \qquad (35)$$

with $\rho_0 = 1000\frac{kg}{m^3}$  unity density.
The spread in deposition velocity for each model is caused mainly by the different wind speeds
during exposure, but also by the variation in relative humidity and, to a lesser extent, by other
thermodynamic conditions. However, it becomes strikingly obvious that in the size range where most
of the atmospheric dust deposition occurs – i.e. between 2 and 50 μm in diameter at Barbados
(Mahowald et al. 2014; van der Does et al. 2016) –, the models disagree by more than two orders of
magnitude. Besides the uncertainty derived from selection of a particular model, the sphericity
assumption and the related drag effects may lead to a bias in deposition flux, most probably mainly
influencing the turbulent deposition regime around 10 μm particle diameter. An additional
measurement bias might be introduced by the parallelism assumption underlying all the stable
boundary layer calculations, i. e. that the air flow must be parallel to the plate. While the vertical



component of the wind speed under atmospheric conditions is usually small in comparison to the
horizontal ones, still 'inlet' losses might occur even due to small non-parallel components. These inlet
losses are expected to affect mainly the largest particles sizes.

### 532   2.4.3   Impactor inlets

The impactor sampler was used with two types of inlets. For particles larger than approximately 2.5
µm aerodynamic diameter, a pseudo-isoaxial inlet orientation with sub-isokinetic sampling was used.
Smaller particles were collected with an omnidirectional inlet. As particles were analyzed separately
for each size class, the inlet efficiency does not play a primary role for the results, still it must be
considered. Literature on an accurate estimation of inlet transmission for a ratio between ambient
wind speed and impactor inlet flow velocity in the range of 100:1 is not existing.  However, from Paik
et al. (2002) and Hangal et al. (1990) in conjunction with the observation of Li et al. (2002) regarding
the applicability of thin walled nozzle formulas to blunt samplers, it may be conclude that:
a) particles larger than 2.5 µm aerodynamic diameter would be increasingly enriched with increasing
particle sizes. Enrichment factors for thin-walled nozzles would be in the range of 2–4 for 10 µm
particles and 20–50 for 100 µm particles. As the sampler had a blunt inlet, the actual enrichment
factors are probably considerably lower.
b) particles smaller than 2.5 µm would be comparatively unbiased at low Stokes numbers; see also
Wen et al. (2000).
For a dry aerosol, these size-selective inlet losses would not considerably bias the relative chemical
composition. In the present humid environment with partly soluble species, though, it can lead to an
overestimation of non-hygroscopic species for particle sizes in the vicinity of the inlet cut-off, if the
hygroscopic growth is not explicitly considered. The problem is somewhat diminished by the fact that
by water-absorption the density of the particles decreases and, consequently, the Stokes number
increases only sub-proportionally to the square of the particle diameter. Nevertheless, the
hygroscopic growth should be explicitly accounted for. Therefore, the model from Eq. (20) is applied
based on the measured geometric diameter and chemical composition, and ambient chemical
compositions are computed.

## 556   2.5   Modelling deposition statistics and artifacts of mixing state

When particles are deposited to a substrate, they might touch each other and form an internal
mixture, which is not representative for the atmosphere. While the lower limit of coincidental
internal particle mixture on a substrate is easily defined – it equals the ratio of the area covered by
particles to the total analyses area for an infinitesimally small depositing particle – the assessment is
much more complex for larger particles following a wide size distribution function.
Therefore, in the first step the deposition process was simulated by a series of Monte Carlo models.
For input, the average size distribution measured at Cape Verde (Kandler et al. 2011b) – hereafter
CV-ground – and the median one measured airborne for aged dust (Weinzierl et al. 2011) – hereafter
CV-air – were used. These size distributions mainly differ in the concentration of supermicron
particles. The deposition velocity formulation after Piskunov (2009) was used. The modeled
deposition area is 5 mm x 5 mm, meteorological conditions were assumed as totally dry, 20 °C, sea
level pressure and a friction velocity of 0.2 m/s. Particles were virtually dropped onto the deposition
surface until either a certain fractional area coverage by particles or a simulated deposition time limit
was reached. Eighteen different area coverages were simulated for a two-component external



mixture (particle density 2200 kg/m³) with components number ratios of 50 % / 50 %, 75 % / 25 %, 90
% / 10 %, 95 % / 5 %, 97 % / 3 %, and 99 % / 1 % for CV-ground, and nine area coverages with number
ratios of 50 % / 50%, 90 % / 10 %, and 99 % / 1 % for CV-air. Each model was run 1000 times (200
times in case of 0.1 and larger fraction area coverages) to assess the statistical uncertainty. In a
second series, for CV-ground a tri-component external mixture of sodium sulfate (particle density
1770 kg/m³), dust (2700 kg/m³) and sea-salt (2170 kg/m³) was used as input. The size-dependent
component number contributions were taken from measurements at Cape Verde (Schladitz et al.
2011). After the simulated deposition, particle agglomerates on the substrate with touching contours
were merged into a new particle with the sum of the volumes and proportionate chemical
composition.
To investigate the relevance of mixing artifacts caused by particle sampling, the sensitivity of
SEM/EDX analysis has to be considered. Internal mixtures can be only detected by SEM/EDX, if the
minor component exceeds the limit of detection. At an acceleration voltage of 12.5 kV the primary X-
ray excitation volume is in the range of 0.5 µm to 1.5 µm diameter, depending on the matrix
elements (Goldstein et al. 2003). As we consider mainly supermicron particles, the excitation volume
is expected to be mainly inside the particles. According to our experience an X-ray peak becomes
detectable at about 0.3 % concentration. Therefore, a 1 % contribution of an element to the particle
volume will be definitely detectable. Thus, a particle containing more than 1 % material from another
particle type is considered as detectable mixture in the model. A particle containing more than 20 %
is denominated as strong internal mixture. Note that for smaller particles, when the excitation
volume would extend into the substrate, larger contributions to the particle volume would be
required.
Besides these fundamental considerations, in the second step a mixing model was applied to each
sample, based on its measured composition. Random particles were virtually selected from the pure
components of the measured set of particles and placed at random positions inside a virtual area
with the same size as the one analyzed in SEM/EDX, until the same area coverage as of the real
sample was reached. Internal mixtures artificially produced on the substrate were counted, if their
mixing would have been detected by SEM/EDX applying the rules for mixed particle classification.
This process was repeated 10,000 times. The upper 95 % confidence interval limit of mixtures
modeled by the Monte Carlo simulation was considered as limit of detection for internal mixtures,
and the median of the produced mixtures was regarded as systematic error and was subtracted from
the mixtures detected in the real samples.
In the third step, the single mixing probability (SMP) for each binary pure compound combination
was calculated by selecting 100,000 random pure-composition particles from the measured data set
for each sample, mixing them virtually and determining, whether they would be detected as mixed.
This was carried out one time without any size restrictions and a second time with only selecting
particles not more than a factor of 3 different in size. The latter was done to account for the fact that
in a turbulent environment and in the regarded size range, the collision efficiency is highest for
particles of similar size (Pinsky et al. 1999; Wang et al. 2005).

### 2.5.1    Simulating particle mixtures due to longer exposure times
While in the modeling section particles are assumed to be spherical, this is typically not the case for
natural aerosol like mineral dust particles. Therefore, a second approach based on particles images
was used to estimate the effect of internal particle mixture on the substrate, i.e. taking into account



the real particle shapes. Due to the large number of images required, this approach could only be
used for assessing the size statistics, but not for the chemical composition. All segmented images of
each deposition sample were subject to particle size analysis. In following steps, a number of 2, 3, 5,
10, 15, or 20 segmented images of the same sample were combined into a single image, simulating
an extension of exposure time by the according factor. This approach inherently assumes a constant
size distribution during exposure and a random particle deposition. The resulting images were then
subject to the same particle analysis, yielding apparent size distributions after a coincidental mixing.
In contrast to the pure modeling approach, here the true size distribution is not known because even
the lowest coverage samples might contain internal mixtures. Certainly though, the lowest coverage
sample is closest to the true size distribution and therefore will be used as reference.

## 3 Results and Discussion

### 3.1 Uncertainty of measurements for the new collection techniques and determination of mixing state

#### 3.1.1 Area homogeneity of collected particles

*Free-wing impactor (FWI)*

To assess the homogeneity of particle distribution, for each sample the center 80 mm² (about 65 % of
the total sample area) were scanned with approximately one thousand SEM images (approximately
one third coverage), and the average particle density was determined for each mm² as function of
$d_g$. For particles between 4 μm and 8 μm in diameter (see Fig. S4 in electronic supplement) no
systematic bias in particle density is visible, except for a slight enhancement toward the borders in a
few cases. The remaining variability remains probably linked to statistical uncertainty and surface
defects interpreted as particles by the automatic segmentation algorithm. However, the density
variations between each mm² remain below a factor of 2. As commonly 20 to 100 mm² are analyzed,
the inhomogeneity can be regarded as minor error. For larger particles, the uncertainty due to
counting statistics becomes dominant.

*Dry particle deposition sampler (DPDS)*

Similar to above, for the DPDS deposition density homogeneity was assessed, but in this case nearly
all of the central 80 mm² were scanned. In about half of the samples, a crescent-shaped density
gradient can be observed (see Fig. S5 in electronic supplement). This gradient most probably
originates from a stationary wave introduced by the recession of the sample substrate slightly below
the primary plane of the DPDS. Depending on the analysis location, a bias in the range of factor 2 to 3
in deposited particle number can occur. Therefore, the fields of analysis for the chemical composition
and size distribution discussion below were homogeneously distributed over the sample surface at a
regular distance. Also with the DPDS, for larger particles the uncertainty due to counting statistics
becomes dominant.

#### 3.1.2 Impact of area coverage and counting statistics on size distribution and total volume

Fig. 6 shows the apparent number and volume size distributions of particles deposited from aerosols
with CV-ground or CV-air size distribution for different area coverages, equaling different exposure
times. As it is to be expected, for short exposure times there is a considerable counting error, which



decreases to less than 10 % for the smaller particles at area coverages of 0.01 and higher. In median,
no particle larger than 50 μm would be detected in deposition area for area coverages smaller than
0.0025, and more than 0.005 are necessary to collect more than 5 particles (not shown in graphs). As
opposing trend there is a bias in size distribution towards lower concentrations and larger particles,
which starts getting relevant at coverages of 0.1. This bias is introduced by the coincidental clumping,
a second particle depositing on an already deposited one. As result, for the given aerosol size
distributions, an area coverage of 0.03 to 0.05 seems most appropriate to get a size distribution
influenced least by counting errors and sampling/mixing bias.
Generally similar, but more pronounced effects can be observed, if the second approach – simulating
longer exposure times by combining real microscope images – is used. Fig. 7 shows for three samples
– low, medium and high area coverages – the evolution of the size distribution due to simulated
longer exposure times. In case of high dust deposition rates and long exposure times, particles
smaller than 10 μm in diameter would be underestimated by a factor of more than 2, while larger
particles would be considerable overrepresented. A shift in the modal diameter of 50 % towards
larger size could be the result. However, at the large end of the volume size distribution, counting
statistics might considerably influence the total particle mass uncertainty, even at these long
simulated exposition times.
If total mass deposition is estimated from the microscope images, one can set up a relation of total
volume and apparent area coverage, which might serve as a quick estimate of total deposited
particle mass (Fig. 8). If the result of the fit function is multiplied with an approximate particle
density, the result gives the deposition as mg/m² with an uncertainty of factor 2. As expected, the fit
function starts to underestimates the volume / mass for high area coverage.
When calculating total mass / volume from small amounts of material, special attention has to be
paid to the errors introduced by counting statistics. Table 1 gives an overview for deposition
simulation results based on a typical area, which would be used for automated single particle
analysis. Two size distributions were considered with different abundance of large particles. Using
the CV-ground size distribution, we observe an uncertainty of a factor of 2 for the total mass (95 %
two-sided confidence interval), when 3,000 particles are counted, which are equivalent to 8 μg of
mass. If only particles between 1 and 32 μm in diameter are regarded, a relative uncertainty of 20 %
is achieved with 1500 particles. When analyzing about 100 μg of particle mass, the statistical error is
in the range of 30 % mass in case of CV-ground size distribution and 15 % for CV-air. It can be
concluded here that a minimum number of 5,000 to 10,000 single particle measurements would be
desirable to stabilize the total mass concentration in the range of 10 % uncertainty. As this number is
usually not reached in SEM studies (e.g., Reid et al. 2003; Coz et al. 2009; Kandler et al. 2011a),
additional attention should be paid to larger particles, e.g. by analyzing larger sample areas, to
decrease the uncertainty in mass (see also Fig. 4). Note that the same considerations in principle
apply to bulk investigations, when only small amounts of mass are analyzed, but are not commonly
stated.
### 3.1.3   Amount of coincidental internal particle mixtures
When assessing the mixing state of particles from an offline single particle technique, coincidental
internal particle mixture has to be taken into account. Fig. 9 shows the upper 95 % confidence limit
of apparent fractions of internally mixed particles for a two-component system as function of source
component ratio and area coverage for detectable strong internal mixtures (refer to section 2.5; data



are given in the electronic supplement, Table S2 and Table S3). These numbers can be considered as
detection limit for fractions of internal mixed particles. As to be expected, higher area coverage
yields higher mixture probability. No significant mixture for submicron particles occurs in these cases.
In particular, if both components are present in equal amounts, mixing probabilities become high
already for covered area fraction of a few percent. Note also the different size maximum for strong
versus detectable mixture.
Applying the same model type based on the CV-ground size distribution to a ternary modal
composition distribution of sulfate, sea-salt and dust as described in section 2.5, mixing probabilities
for a specific atmospheric composition can be estimated (Fig. 10). Note the different color bar scale.
It becomes instantly obvious that the mixing probabilities are much lower than in the homogeneous
case. Mixtures between sulfate and sea-salt as well as ternary mixture are absent. The relative
fraction of internally mixed particles is lower by an order of magnitude. This can be explained by the
fact that the defined relative detection limits of 20 % and 1 % restrict the detection of mixing to
mixing partners not differing in size by more than a factor of 1.59 (strong mixing) and 4.6 (detectable
mixing). But because different aerosol type are mainly present in different size regimes here
(Schladitz et al. 2011), the mixture can only be efficient for size ranges, where these component have
an overlap. In general, however, also here mixture increases with particle size.
It can be concluded here that mixing studies for large particles are generally very difficult. Many
particles need to be collected in total to ensure reliable counting statistics, which leads in
consequence to high mixing probabilities. This issue is of less concern for particles smaller than 10
µm for the given size distributions and in cases, where the aerosol has a strong dependence of
composition on particle size. It also emphasizes that mixing studies should be accompanied by
mixture modeling as performed below.

## 3.2 Field Measurements – methodical aspect

### 3.2.1 Comparison of atmospheric size and volume concentrations

Using the FWI sampling efficiencies outlined in section 2.3.3 and the DPDS deposition velocities from
2.4.1, one can calculate the atmospheric size distribution derived by the two techniques. Fig. 11
shows the average size distributions for the post- and pre-storm periods based on different
deposition velocity models for total and upper estimate dust mass concentrations. The lower dust
estimate (not shown) exhibits qualitatively the same behavior. It is evident that there is a large
discrepancy between the different models as well as between the DPDS and FWI measurements. The
discrepancy is clearly larger than the statistical uncertainties. While the total mass median diameter
derived from DPDS (Piskunov model) is around 5 µm particle diameter, for the FWI it is
approximately 25 µm. A dust size distribution measured in the Saharan Air Layer in 2.3 km altitude
(computed from data shown by Weinzierl et al. 2017) contains the same mode around 4 µm
diameter, but shows a secondary maximum at 10 µm, which is not found by the ground-based
measurements. It is interesting to note that these values get closer, when only the dust fraction is
considered, indicating a connection of the discrepancy with the hygroscopic growth (e.g., growth or
density misestimate). Two other reasons for the discrepancy might be for the FWI an uncertain
collection efficiency and particle losses due to non-parallel flow for the DPDS. The FWI has 50 %
collection efficiency around 11 µm particle aerodynamic diameter, so for smaller particles – the
majority by far – the efficiency correction function may yield unrealistic values. The DPDS model
assumptions require a well oriented flow. At the high wind speeds, a non-zero angle-of-attack flow





(from below) might lead to considerable particle losses for the larger particles. This might for
example be caused by an increased boundary layer thickness over the lower plate. Such an angular
flow was observed at the measurement site due to the cape orography.
When total mass is calculated from deposition, it can be compared to dust concentration
measurements with a high volume filter sampler. Fig. 12 shows time series of mass concentrations
measured by the high-volume sampler, estimated from dry deposition measurements as well as the
raw dry deposition flux densities. For dry deposition uncertainties derived from the low / upper
estimates as well as from counting statistics are shown. A few things can be learned from this data.
With respect to the deposition model, the Piskunov model performs rather well. The average of the
high-volume sampler mass concentration time series (see Table 2) is close to the lower estimate of
the Piskunov model, while the higher estimate overestimates the mass concentration. The other
models deviate considerably more, as to be expected from the deposition velocity differences (Fig.
5). The ratio of the mass concentration estimate to the mass flux density varies over slightly more
than one order of magnitude, depending mainly on size distribution and wind conditions. High
volume and deposition-estimated mass concentrations as well as the mass flux densities follow
qualitatively the same pattern in showing low concentration and high concentration periods. The
absolute numbers, however, deviate significantly. For sub-periods, the correlation quality seems to
be different. E.g., starting from June 21, the correlation of mass flux with high volume mass
concentrations seems to be better than the one with deposition estimated concentrations; for the
period before June 21 situation is converse. No direct link of the correlations with any meteorological
variable was found, indicating that the deviations depend in part on erroneous assumptions in the
model. For example, tuning other deposition velocity models by arbitrary factors can lead to a better
agreement of actively and passively determined mass concentrations for this particular data set (Fig.
S9 in electronic supplement), but the data basis is too small for a robust tuning without physical
backing. Moreover, disagreement might also be caused by physical measurement biases like
unknown size-dependent inlet efficiency for the high-volume sampler or angular inflow for the DPDS.

## 3.3   Field Measurements – atmospheric and aerosol aspects

### 3.3.1   Aerosol composition

Overall aerosol composition (i.e. the relative number abundance of the different particle groups) was
measured by electron microscopy single particle analysis (Fig. 13). The relative abundance of soluble
sulfate is highest for the smallest particle sizes, which is in good accordance with previous
measurements in the eastern Atlantic Ocean (Kandler et al. 2011a). After the storm passage, higher
sulfate abundances – soluble as well as stable – are observed in 2013, which are similar to those
observed in 2016. The sea-salt abundance is higher for the pre-storm period in 2013, which is in
agreement with the wind speeds observed (see below). In 2016, a much higher abundance of small
Fe-rich particles (contained in the oxides/hydroxides class) is observed compared to the pre-storm
period in 2013. For the post-storm period in 2013, minor amounts of these particles are visible.
Overall, an average dust deposition of 10 mg m$^{-2}$ d$^{-1}$ (range 0.5–47 mg m$^{-2}$ d$^{-1}$) is observed (Fig. 14).
While a strict disambiguation can't be done elements also found in sea-salt, Al, Si ,P, Ti, and Fe are
most like derived from dust only and are therefore also shown in the graph. At Barbados, Fe
contributes 0.67 (0.01–3.3) mg m$^{-2}$ d$^{-1}$ to deposition, while phosphorous adds only 0.001 mg m$^{-2}$ d$^{-1}$;
however, P is below the detection limit on two thirds of the days. The cumulative size distribution
shows that in particular P and Ti are located preferentially within smaller particles. Al, Si and Fe show



generally a similar size distribution. As corroborated by the results above, Fe is slightly more present
in particles smaller than 5 µm, but the impact of these periods with small Fe-rich particles on total Fe
deposition is obviously small, owing to the lower overall deposition rate during these periods.
**3.3.2    Airmass history and potential aerosol sources**
The airmass provenance of the sampling periods in 2013 and 2016 is generally similar. The
trajectories mostly followed the trade-wind path from North-West Africa and East Atlantic Ocean to
Barbados (Fig. S6 in electronic supplement). In 2013, the air was coming more frequently from
Western African than in 2016. After the tropical storm Chantal in 2013, the airmass origin shifted
slightly to more southern regions. In a few cases in 2013, air from the North-West Atlantic Ocean was
recirculated into the trade-wind path. In 2016, airmasses from North-East Southern America were
more frequent than in 2016.
The sea-salt deposition rates are not linked to air mass provenance (not shown). The dust
provenance for both years (Fig. 15) is – as expected – pointing to West Africa. This source region is
also identified by isotope measurements in July/August 2013 (Bozlaker et al. 2018). The soluble
sulfate deposition is generally linked to three regions, the Atlantic Ocean, West Africa and south west
Europe. In particular in 2016, the sulfate sources appear to be located more in Europe and less in
Africa. The relative ion balance shows mostly slightly negative values indicating presence of $NH_4^+$ or
$H^+$. Interestingly, a positive ion excess is observed for European sulfate in 2016, indicating possible
presence of $NO_3^-$. These observations support the hypothesis that sulfate associated with dust events
at Barbados partly might originate from secondary processing of European precursors (Li-Jones et al.
1998).

Iron contribution from dust is of particular interest for marine ecosystems. Therefore, Fig. 16 shows
in the upper panel the silicate SIAI as proxy for quick iron availability. It is obvious, that the iron-
containing silicate particle source is located in West Africa. Northern and southern West Africa as
source regions can't be distinguished after trans-Atlantic transport, in contrast to investigations close
to the source (Kandler et al. 2007). This is consistent with observations based on isotope analysis,
where also a homogeneous composition has been observed at Barbados (Bozlaker et al. 2018). In
comparison with Fig. 15, a slightly higher SIAI can be observed in 2016 than in 2013, while the dust
deposition rates in contrast are lower. While the total iron deposition correlates well with dust
deposition (not shown), similar to observations by Trapp et al. (2010), for the SIAI an inverse
relationship is found at Barbados with higher dust deposition rates leading to lower ratios of SIAI to
total dust. This correlates to previous findings, where iron solubility decreased with increasing dust
concentration (Shi et al. 2011b; Sholkovitz et al. 2012), though no direct causal relationship can be
derived (Shi et al. 2011a). As the finding indicates a higher iron contribution from smaller particles,
the lower row of Fig. 16 shows the Fe contribution by small iron-rich grains. While in 2013 the
contribution of this particle type is generally low, in 2016, when trajectories cross the North-Eastern
tip of South America, there is a low input during low-dust situations. However, trajectories arriving
straight from south east would also cross Barbados before arrival, so there might also local
contribution from the island. Taking a closer look at these particles (Fig. S7 in the electronic
supplement) it reveals that they differ in structure from usual mineral dust particles (Moreno et al.
2006; Scheuvens et al. 2011; Deboudt et al. 2012), but resemble more closely material from
industrial or combustion processes (Fu et al. 2014; Hu et al. 2015; Li et al. 2016). This observation is
interpreted as evidence for an anthropogenic iron input into a marine environment. This is of



particular interest, as according to previous work, the bioavailability of anthropogenic iron is usually
higher than provided by natural sources (Desboeufs et al. 2005; Sedwick et al. 2007; Fu et al. 2014).

### 3.3.3  Sea-salt composition

When considering sea-salt composition, it is assumed generally that except from the sulfate content,
aerosol produced from sea-water has a major composition resembling the bulk sea-water (Lewis et
al. 2004). However, it was recently shown in the Arctic that a fractionation can occur also with
respect to the major composition (Salter et al. 2016). At Barbados, an increasing positive deviation
from the nominal value of 0.022 with decreasing particle size is observed for the Ca/Na atomic ratio
of sea-salt particles (Fig. 17). This indicates that the same effects found by Salter et al. (2016) are
present in Caribbean sea-salt production. According to the authors, these might be linked to an
enrichment of Ca in sea surface micro-layers, but details are not yet known.

### 3.3.4  Abundance of mixed particles

If we consider the abundance of mixed particles at Barbados, a complex picture emerges as function
of particle size, time period and available mixing partners (Fig. 18). It can be observed that the total
deposition rate for all particle types is linked to the wind speed, what is to be expected from the
physical process (see for example Fig. S8 in electronic supplement). The higher sea-salt deposition
rates and also higher concentrations in 2013 in comparison to 2016 are also linked to the wind
speed, showing the local sea-salt production. In contrast, the dust concentration is slightly lower for
higher wind speeds (Fig. S8) for both years. With increasing particle size, the relative abundance of
internal dust/sea-salt mixtures increases (Fig. 18), but these mixtures only occur when considerable
amounts of sea-salt are present. This is different for the internal mixture with sulfate. While there
are similar ratios of dust and sulfate particles observed in the second half of the 2013 data as in
2016, in 2013, dust/sulfate mixtures are practically absent. Assuming that higher wind speeds in 2013
should lead to more internal mixing due to increased turbulence, this is clearly indicating that in
contrast to the sea-salt/dust mixture, the sulfate/dust mixture has a non-local origin (e.g., Usher et
al. 2002).
This is corroborated by the dependence of internal mixtures relative abundance on the single mixing
probability (Fig. S10 in the electronic supplement). If one considers here the binary number fraction
of mixed particles – i.e. ratio of binary mixed particles to pure compounds – as function of the size-
restricted single mixing probability, there is a weak positive correlation for dust/sea-salt mixtures for
particles larger than 2 μm diameter, but no correlation for dust/sulfate mixtures. Moreover, for
similar single mixing probabilities, the binary number fraction of mixed particles appears slightly
higher for higher deposition rates. As the collision efficiency depends on the square of the number
concentration (Sundaram et al. 1997), this supports the hypothesis of a locally produced internal
mixture of sea-salt and dust and a non-local production of sulfate and dust, the latter having most
probably cloud processing involved (Andreae et al. 1986; Niimura et al. 1998).
The overall ratio of dust/sea-salt internal mixture abundance to all dust- and sea-salt-particles
increases from 0.01–0.03 for 1 μm particles to 0.1–0.7 for particles of 8–16 μm in diameter, whereas
for dust/sulfate mixtures the ratio of 0.01–0.02 is not dependent on particle size. Denjean et al.
(2015) report mixed particle abundances of 0.16–0.3, but do not state a size range, so the data can't
be compared directly.



If the findings on Barbados are compared to measurements in the eastern Atlantic Ocean (Kandler et al. 2011a), a generally lower abundance of internally mixed particles with respect to dust/sulfate is observed, while comparable abundances of sea-salt/dust mixtures are found. While the latter can be explained by similar wind conditions and comparable single mixing probabilities, the former seems to be caused by different aging conditions. Dust arriving over Barbados is transported mostly in the dry Saharan Air Layer (e.g., Schütz 1980), while dust arriving during winter-time at Cape Verde is transported inside the humid marine boundary layer (Chiapello et al. 1995; Kandler et al. 2011b). Therefore, considerably higher chemical processing rates at Cape Verde due to the higher humidity can be expected (Dlugi et al. 1981; Ullerstam et al. 2002), even though the transport time is most likely shorter. In addition, the boundary layer most probably provides higher concentrations of sulfur compounds for reaction (Davison et al. 1996; Andreae et al. 2000).

***Change in dust behavior due to internal particle mixing***

If dust particles become internally mixed, their mass, size and hygroscopic behavior change. Therefore, they will have modified deposition velocities as well as hygroscopic properties. Fig. 19 shows the increases in deposition velocities for mixed particles observed at Ragged Point in 2013 and 2016. For the both mixtures (dust/sea-salt and dust/sulfate), an increase at ambient conditions of a factor of 2–3 is observed for submicron dust particles, which rises to a factor of 5–10 for particles of 3 μm dust core diameter. As a result, the dust average deposition velocity for particles between 1 and 10 μm aerodynamic diameter is increased by 30–140 % at ambient conditions (Fig. 20). Considering a mass mean aerodynamic diameter in deposition of 7.0 μm, at ambient conditions dust deposition velocity is 6.4 mm/s, which is an enhancement by approximately 35 % over the unmixed state. This overall value is in the range estimated by Prospero et al. (2009). The enhancement will become more pronounced at higher humidities. It has to be emphasized that this estimate is a lower limit, as there most likely exist mixed particles with a smaller contribution of hygroscopic material, which remaining undetected by our analytical approach. At higher humidities, this smaller contribution nevertheless will increase the deposition velocity of the mixtures. While we observe similar relative abundances of mixed particles to previous work in Asian dust outflow Zhang (2008), our estimate of impact on deposition is considerable higher, which is mainly related to the use of the Piskunov model taking into account turbulent deposition over a Stokes settling approach.

An internal mixture of dust with a soluble compound will also modify the in-cloud behavior of the dust particles. As Denjean et al. (2015) have shown, in the Caribbean only dust particles internally mixed with soluble compounds exhibit considerable hygroscopic growth. Therefore, if the mixed dust particles are entrained into a cloud, they would preferentially be activated into a cloud droplet in comparison to unmixed dust. This according cloud droplets would then contain a potential effective ice-nucleating particle (DeMott et al. 2003). Because the ice-nucleating efficiency of a mixed droplet would follow the most efficient compound (Augustin-Bauditz et al. 2016), these droplets would at according temperatures become ice particles by immersion freezing (Marcolli et al. 2007; Niemand et al. 2012). We may hypothesize that along this path – by internal mixture with soluble compounds – the atmospheric ice-nucleating efficiency of dust particles could be enhanced. For example, within mixed-phase clouds at a free-troposphere station, high abundances of internally mixed particles were found as ice-nucleating particles or ice particle residues (Ebert et al. 2011; Worringen et al. 2015). Note that this mixing path is probably restricted to immersion freezing (Eastwood et al. 2009).





## 4   Summary and Conclusions

Aerosol deposition measurements by means of passive samplers were carried out on a daily basis at Ragged Point, Barbados in June/July 2013 and August 2016. In addition, active aerosol collection was performed with a cascade and a novel free-wing impactor. Size, shape and composition of about 110,000 particles were determined by electron microscopy. Focus was placed in this work on measurement accuracy of chemical composition and mixing state determination for individual particles.

Ragged Point, in particular in 2013, is a high-wind and high-humidity environment, which considerably influences representativeness and accuracy of the different sampling techniques. A deposition model including chemistry-dependent hygroscopic growth was adapted to the sampling situation to assess atmospheric concentration of large particles. Fair agreement was reached between passive and active techniques regarding mass concentration, but clear discrepancies were observed for particle size distribution.

Special attention was paid to the mixing state of dust particles.  A model was developed to assess the mixing state of airborne particles by correcting for sampling artifacts due to particle overload leading to coincidental internal mixing of particles on the substrate (i.e., not representative for the airborne state). Different approaches were tested based on model size distributions and observed particle deposition images. It was found that the size distribution is only weakly affected for substrate area coverages with particles below 10 %. The chemical composition of mixtures, however, is already affected at much lower area coverages of < 1 %.

During our measurement campaigns, the aerosol was dominated by dust, sea-salt and sulfate in changing proportions. The sea-salt concentration at Ragged Point is mainly depending on wind speed. Back trajectory analysis showed that dust is originating from the usual sources in West Africa. Sulfate showed three major potential source areas, Africa, Europe and Atlantic Ocean. Particularly in 2013, sulfate was more linked to the African source, while in 2016 southwest Europe occurred as potential source, with a possible contribution of nitrate. In 2016 for short time periods, contributions to iron deposition from probably anthropogenic sources (potentially as magnetite) from South America or the island of Barbados were observed.

It was further found that internal mixing of dust and sea-salt is depending on local wind speed, and we, thus, hypothesize that it is produced locally, most likely by turbulent processes. In contrast, mixtures of dust and soluble sulfates are presumably not produced locally, but may have formed during the inter-continental transport. Even though the overall amount of internally mixed particles is comparatively low, a considerable impact on total dust deposition velocity is estimated. In addition, a pathway is hypothesized by which the ice-nucleation efficiency of dust can be increased by mixing with soluble compounds during or after the long-range transport.

For future work, some conclusions can be drawn from our observations:

- If different techniques for deposition and/or atmospheric concentration measurements are compared, it is crucial to measure particles size distributions. We observed in some cases that total mass concentration can compare rather well, even though size distributions – and therefore collection efficiencies – are considerably different.



- A better understanding – in theory as well as in experimental use – of particle deposition and collection efficiencies is required in particular under high wind-situations, where turbulent transport has a considerable impact. This most probably applies to a wide range of deposition samplers, not only these used in this work.
- When mixing state investigations are done based on collected aerosol particles, the impact of coincidental mixtures on the substrate must be assessed, unless the area coverage with particles is very low (<< 1%). This is particularly the case for larger particles (> 5 µm diameter) and for aerosols in the same size range, where similar abundances of different mixing partners exist.
- Internal particle mixing most likely has a considerable influence on dust deposition speed and on the impact of dust on clouds. Future models regarding dust deposition should take a deposition speed enhancement by internal mixing into account. However, more systematic investigations are needed to better understand the mixing processes.
- With respect to the cloud impact if mixing via a more efficient incorporation of immersion freezing ice nuclei into cloud droplets by preferential activation, future ice nucleation chamber experiments are needed to assess the importance of this effect.
- Finally, a larger data basis beyond the observation of single events is required to assess the anthropogenic influence on the iron deposition into the Oceans, besides the input by mineral dust. This data base needs to be acquired with a high time resolution (maximum days) to match the duration of the observed deposition events. Also, these time series would need to cover months to years at Barbados, given the probably infrequent occurrence of these events.

## 5  Data availability

The data sets of all particles used for this investigation including particle size, shape, and composition are given as text tables in the electronic supplement along with a data overview.

## 6  Author contribution

KK designed the experiment. KK and MH carried out field work in 2013. MP and CP carried out the field work in 2016. KK and KS analyzed the samples. KK programmed the models and data processing code. KK, SW and ME analyzed data and prepared the manuscript. All authors contributed in data discussion and manuscript finalization.

## 7  Competing interests

The authors declare that they have no conflict of interest.

## 8  Acknowledgements

We acknowledge financial support from the German Research foundation (DFG grant KA 2280/2-1 and KA 2280/3-1). We thank Joseph Prospero for his valuable comments on the manuscript and discussion; his wind and mass concentration data were obtained under National Science Foundation (NSF) grant AGS-0962256. The authors gratefully acknowledge the NOAA Air Resources Laboratory



(ARL) for the provision of the HYSPLIT transport and dispersion model and/or READY website
(http://www.ready.noaa.gov) used in this publication.

# 9  Literature

Agresti, A., B. A. Coull (1998): Approximate Is Better than "Exact" for Interval Estimation of Binomial
Proportions. Am. Stat. 52(2), 119-126. doi: 10.2307/2685469
Aluko, O., K. E. Noll (2006): Deposition and Suspension of Large, Airborne Particles. Aerosol Sci.
Technol. 40(7), 503-513.
Andreae, M. O. (1995). Climatic effects of changing atmospheric aerosol levels. Future climates of the
world: a modelling perspective. A. Henderson-Sellers. Amsterdam, The Netherlands, Elsevier.
**16:** 347-398.
Andreae, M. O., R. J. Charlson, F. Bruynseels, H. Storms, R. V. Grieken, W. Maenhaut (1986): Internal
Mixture of Sea Salt, Silicates, and Excess Sulfate in Marine Aerosols. Science 232, 1620-1623.
Andreae, M. O., W. Elbert, R. Gabriel, D. W. Johnson, S. Osborne, R. Wood (2000): Soluble ion
chemistry of the atmospheric aerosol and SO2 concentrations over the eastern North
Atlantic during ACE-2. Tellus B 52(4), 1066-1087. doi: 10.3402/tellusb.v52i4.17087
Armstrong, J. T. (1991). Quantitative elemental analysis of individual microparticles with electron
beam instruments. Electron probe quantitation. K. F. J. Heinrich and D. E. Newbury. New
York, Plenum Press**:** 261-315.
Ashbaugh, L. L., W. C. Malm, W. Z. Sadeh (1985): A residence time probability analysis of sulfur
concentrations at grand Canyon National Park. Atmos. Environ. 19(8), 1263-1270. doi:
10.1016/0004-6981(85)90256-2

Augustin-Bauditz, S., H. Wex, C. Denjean, S. Hartmann, J. Schneider, S. Schmidt, M. Ebert, F.
Stratmann (2016): Laboratory-generated mixtures of mineral dust particles with biological
substances: characterization of the particle mixing state and immersion freezing behavior.
Atmos. Chem. Phys. 16(9), 5531-5543. doi: 10.5194/acp-16-5531-2016
Baker, A. R., T. D. Jickells (2006): Mineral particle size as a control on aerosol iron solubility.
Geophysical Research Letters 33(17), L17608. doi: 10.1029/2006gl026557
Bozlaker, A., J. M. Prospero, J. Price, S. Chellam (2018): Linking Barbados Mineral Dust Aerosols to
North African Sources Using Elemental Composition and Radiogenic Sr, Nd, and Pb Isotope
Signatures. J. Geophys. Res. 123(2), 1384-1400. doi: doi:10.1002/2017JD027505
Carpenter, J., J. Bithell (2000): Bootstrap confidence intervals: when, which, what? A practical guide
for medical statisticians. Stat. Med. 19(9), 1141-1164. doi: 10.1002/(sici)1097-
0258(20000515)19:9<1141::aid-sim479>3.0.co;2-f
Chiapello, I., G. Bergametti, L. Gomes, B. Chatenet, F. Dulac, J. Pimenta, E. Santos Suares (1995): An
additional low layer transport of Sahelian and Saharan dust over the North-Eastern Tropical
Atlantic. Geophys. Res. Lett. 22(23), 3191-3194. doi: 10.1029/95GL03313
Choobari, O. A., P. Zawar-Reza, A. Sturman (2014): The global distribution of mineral dust and its
impacts on the climate system: A review. Atmos. Res. 138(0), 152-165. doi:
10.1016/j.atmosres.2013.11.007
Cliff, G., G. W. Lorimer (1975): The quantitative analysis of thin specimens. J. Microsc. 103(2), 203-
207. doi: 10.1111/j.1365-2818.1975.tb03895.x
Coz, E., F. J. Gómez-Moreno, M. Pujadas, G. S. Casuccio, T. L. Lersch, B. Artíñano (2009): Individual
particle characteristics of North African dust under different long-range transport scenarios.
Atmos. Environ. 43, 1850-1863.
Dall'Osto, M., R. M. Harrison, E. J. Highwood, C. O'Dowd, D. Ceburnis, X. Querol, E. P. Achterberg
(2010): Variation of the mixing state of Saharan dust particles with atmospheric transport.
Atmos. Environ. 44(26), 3135-3146. doi: 10.1016/j.atmosenv.2010.05.030
Davison, B., C. O'Dowd, C. N. Hewitt, M. H. Smith, R. M. Harrison, D. A. Peel, E. Wolf, R. Mulvaney, M.
Schwikowski, U. Baltensperger (1996): Dimethyl sulfide and its oxidation products in the





atmosphere of the Atlantic and Southern Oceans. Atmos. Environ. 30(10), 1895-1906. doi: 10.1016/1352-2310(95)00428-9

Deboudt, K., P. Flament, M. Choël, A. Gloter, S. Sobanska, C. Colliex (2010): Mixing state of aerosols and direct observation of carbonaceous and marine coatings on African dust by individual particle analysis. J. Geophys. Res. 115, D24207. doi: 10.1029/2010JD013921

Deboudt, K., A. Gloter, A. Mussi, P. Flament (2012): Red-ox speciation and mixing state of iron in individual African dust particles. J. Geophys. Res. 117, D12307. doi: 10.1029/2011JD017298

Deer, W. A., R. A. Howie, J. Zussman (1992): An Introduction to the Rock-Forming Minerals. Second Edition. Harlow, UK, Pearson Education Ltd.

DeMott, P., K. Sassen, M. Poellot, D. Baumgardner, D. Rogers, S. Brooks, A. Prenni, S. Kreidenweis (2003): African dust aerosols as atmospheric ice nuclei. Geophys. Res. Lett. 30, 1732. doi: 10.1029/2003GL017410

Denjean, C., S. Caquineau, K. Desboeufs, B. Laurent, M. Maille, M. Quiñones Rosado, P. Vallejo, O. L. Mayol-Bracero, P. Formenti (2015): Long-range transport across the Atlantic in summertime does not enhance the hygroscopicity of African mineral dust. Geophys. Res. Lett. 42(18), 7835-7843. doi: 10.1002/2015gl065693

Desboeufs, K. V., A. Sofikitis, R. Losno, J. L. Colin, P. Ausset (2005): Dissolution and solubility of trace metals from natural and anthropogenic aerosol particulate matter. Chemosphere 58(2), 195-203. doi: 10.1016/j.chemosphere.2004.02.025

DiCiccio, T. J., B. Efron (1996): Bootstrap confidence intervals. Statist. Sci. 11(3), 189-228. doi: 10.1214/ss/1032280214

Dlugi, R., S. Jordan, E. Lindemann (1981): The heterogeneous formation of sulfate aerosols in the atmosphere. J. Aerosol Sci. 12(3), 185-197. doi: 10.1016/0021-8502(81)90089-6

Eastwood, M. L., S. Cremel, M. Wheeler, B. J. Murray, E. Girard, A. K. Bertram (2009): Effects of sulfuric acid and ammonium sulfate coatings on the ice nucleation properties of kaolinite particles. Geophys. Res. Lett. 36, L02811.

Ebert, M., A. Worringen, N. Benker, S. Mertes, E. Weingartner, S. Weinbruch (2011): Chemical composition and mixing-state of ice residuals sampled within mixed phase clouds. Atmos. Chem. Phys. 11, 2805-2816. doi: 10.5194/acp-11-2805-2011

Efron, B. (1979): Bootstrap Methods: Another Look at the Jackknife. Ann. Statist.(1), 1-26. doi: 10.1214/aos/1176344552

Efron, B. (2003): Second Thoughts on the Bootstrap. Statist. Sci. 18(2), 135-140. doi: 10.1214/ss/1063994968

Eglinton, T. I., G. Eglinton, L. Dupont, E. R. Sholkovitz, D. Montluçon, C. M. Reddy (2002): Composition, age, and provenance of organic matter in NW African dust over the Atlantic Ocean. Geochem. Geophy. Geosy. 3(8), 1-27. doi: 10.1029/2001gc000269

Falkovich, A. H., E. Ganor, Z. Levin, P. Formenti, Y. Rudich (2001): Chemical and mineralogical analysis of individual mineral dust particles. J. Geophys. Res. 106(D16), 18029-18036.

Fitzgerald, E., A. P. Ault, M. D. Zauscher, O. L. Mayol-Bracero, K. A. Prather (2015): Comparison of the mixing state of long-range transported Asian and African mineral dust. Atmos. Environ. 115, 19-25. doi: 10.1016/j.atmosenv.2015.04.031

Fletcher, R. A., N. W. M. Ritchie, I. M. Anderson, J. A. Small (2011). Microscopy and microanalysis of individual collected particles. Aerosol Measurement. Principles, Techniques, and Applications. P. Kulkarni, P. A. Baron and K. Willeke. Hoboken, New Jersey, John Wiley & Sons.: 179-232.

Fu, H. B., G. F. Shang, J. Lin, Y. J. Hu, Q. Q. Hu, L. Guo, Y. C. Zhang, J. M. Chen (2014): Fractional iron solubility of aerosol particles enhanced by biomass burning and ship emission in Shanghai, East China. Sci. Total Environ. 481, 377-391. doi: 10.1016/j.scitotenv.2014.01.118

Goldstein, J. I., D. E. Newbury, D. Joy, C. Lyman, P. Echlin, E. Lifshin, L. Sawyer, J. Michael (2003): Scanning Electron Microscopy and X-Ray Microanalysis. New York, Kluwer Academic / Plenum Publishers.



Golovin, M. N., A. A. Putnam (1962): Inertial impaction on single elements. Ind. Eng. Chem. Fundam. 1(4), 264-273.

Grini, A., G. Myhre, C. S. Zender, I. S. A. Isaksen (2005): Model simulations of dust sources and transport in the global atmosphere: Effects of soil erodibility and wind speed variability. J. Geophys. Res. 110, 10.1029/2004JD005037.

Hangal, S., K. Willeke (1990): Overall efficiency of tubular inlets sampling at 0 - 90 degrees from horizontal aerosol flows. Atmos. Environ. 24A, 2379-2386. doi: 10.1016/0960-1686(90)90330-P

Hartung, J., B. Elpelt, K.-H. Klösener (2005): Statistik: Lehr- und Handbuch der angewandten Statistik., Oldenbourg, Munich, Germany.

Hinds, W. C. (1999): Aerosol Technology. Properties, behavior, and measurement of airborne particles. Second edition. New York, USA, Wiley Interscience.

Hu, Y., J. Lin, S. Zhang, L. Kong, H. Fu, J. Chen (2015): Identification of the typical metal particles among haze, fog, and clear episodes in the Beijing atmosphere. Sci. Total Environ. 511, 369-380. doi: 10.1016/j.scitotenv.2014.12.071

Jaenicke, R., C. Junge (1967): Studien zur oberen Grenzgröße des natürlichen Aerosols. Beitr. Phys. Atmos. / Contrib. Atmos. Phys. 40, 129-143.

Kandler, K., N. Benker, U. Bundke, E. Cuevas, M. Ebert, P. Knippertz, S. Rodríguez, L. Schütz, S. Weinbruch (2007): Chemical composition and complex refractive index of Saharan Mineral Dust at Izaña, Tenerife (Spain) derived by electron microscopy. Atmos. Environ. 41(37), 8058-8074. doi: 10.1016/j.atmosenv.2007.06.047

Kandler, K., K. Lieke, N. Benker, C. Emmel, M. Küpper, D. Müller-Ebert, M. Ebert, D. Scheuvens, A. Schladitz, L. Schütz, S. Weinbruch (2011a): Electron microscopy of particles collected at Praia, Cape Verde, during the Saharan Mineral dust experiment: particle chemistry, shape, mixing state and complex refractive index. Tellus 63B, 475-496. doi: 10.1111/j.1600-0889.2011.00550.x

Kandler, K., L. Schütz, C. Deutscher, H. Hofmann, S. Jäckel, P. Knippertz, K. Lieke, A. Massling, A. Schladitz, B. Weinzierl, S. Zorn, M. Ebert, R. Jaenicke, A. Petzold, S. Weinbruch (2009): Size distribution, mass concentration, chemical and mineralogical composition, and derived optical parameters of the boundary layer aerosol at Tinfou, Morocco, during SAMUM 2006. Tellus 61B, 32-50. doi: 10.1111/j.1600-0889.2008.00385.x

Kandler, K., L. Schütz, S. Jäckel, K. Lieke, C. Emmel, D. Müller-Ebert, M. Ebert, D. Scheuvens, A. Schladitz, B. Šegvić, A. Wiedensohler, S. Weinbruch (2011b): Ground-based off-line aerosol measurements at Praia, Cape Verde, during the Saharan Mineral Dust Experiment: Microphysical properties and mineralogy. Tellus 63B, 459-474. doi: 10.1111/j.1600-0889.2011.00546.x

Karyampudi, V. M., S. P. Palm, J. A. Reagen, H. Fang, W. B. Grant, R. M. Hoff, C. Moulin, H. F. Pierce, O. Torres, E. Browell, S. H. Melfi (1999): Validation of the Saharan Dust Plume Conceptual Model Using Lidar, Meteosat, and ECMWF Data. Bull. Am. Met. Soc. 80(6), 1045-1075.

Karydis, V. A., A. P. Tsimpidi, S. Bacer, A. Pozzer, A. Nenes, J. Lelieveld (2017): Global impact of mineral dust on cloud droplet number concentration. Atmos. Chem. Phys. 17(9), 5601-5621. doi: 10.5194/acp-17-5601-2017

Koehler, K. A., P. J. DeMott, S. M. Kreidenweis, O. B. Popovicheva, M. D. Petters, C. M. Carrico, E. D. Kireeva, T. D. Khokhlova, N. K. Shonija (2009): Cloud condensation nuclei and ice nucleation activity of hydrophobic and hydrophilic soot particles. Phys. Chem. Chem. Phys. 11, 7906-7920. doi: 10.1039/b905334b

Kristensen, T. B., T. Müller, K. Kandler, N. Benker, M. Hartmann, J. M. Prospero, A. Wiedensohler, F. Stratmann (2016): Properties of cloud condensation nuclei (CCN) in the trade wind marine boundary layer of the western North Atlantic. Atmos. Chem. Phys. 16(4), 2675-2688. doi: 10.5194/acp-16-2675-2016





Lai, A. C. K., W. W. Nazaroff (2005): Supermicron particle deposition from turbulent chamber flow onto smooth and rough vertical surfaces. Atmos. Environ. 39(27), 4893-4900. doi: 10.1016/j.atmosenv.2005.04.036

Laskin, A., J. P. Cowin, M. J. Iedema (2006): Analysis of individual environmental particles using modern methods of electron microscopy and X-ray microanalysis. J. Electron. Spectrosc. Relat. Phenom. 150, 260-274. doi: 10.1016/j.elspec.2005.06.008

Lepple, F. K., C. J. Brine (1976): Organic constituents in eolian dust and surface sediments srom northwest Africa. J. Geophys. Res. 81(6), 1141-1147. doi: 10.1029/JC081i006p01141

Lewis, E. R., S. E. Schwartz (2004): Sea Salt Aerosol Production: Mechanisms, Methods, Measurements and Models. Washington, DC, American Geophysical Union.

Li-Jones, X., J. M. Prospero (1998): Variations in the size distribution of non-sea-salt sulfate aerosol in the marine boundary layer at Barbados: Impact of African dust. J. Geophys. Res. 103(D13), 16073-16084. doi: 10.1029/98jd00883

Li, S.-N., D. A. Lundgren (2002): Aerosol Aspiration Efficiency of Blunt and Thin-Walled Samplers at Different Wind Orientations. Aerosol Sci. Tech. 36, 342-350.

Li, W., L. Shao, D. Zhang, C.-U. Ro, M. Hu, X. Bi, H. Geng, A. Matsuki, H. Niu, J. Chen (2016): A review of single aerosol particle studies in the atmosphere of East Asia: morphology, mixing state, source, and heterogeneous reactions. J. Clean. Prod. 112, 1330-1349. doi: 10.1016/j.jclepro.2015.04.050

Liao, H., J. H. Seinfeld (1998): Radiative forcing by mineral dust aerosols: sensitivity to key variables. J. Geophys. Res. 103(D24), 31637-31645. doi: 10.1029/1998JD200036

Liu, B. Y. H., J. K. Agarwal (1974): Experimental observation of aerosol deposition in turbulent flow. J. Aerosol Sci. 5(2), 145-155. doi: 10.1016/0021-8502(74)90046-9

Mahowald, N., S. Albania, J. F. Kok, S. Engelstaeder, R. Scanza, D. S. Ward, M. G. Flanner (2014): The size distribution of desert dust aerosols and its impact on the Earth system. Aeolian Res. 15, 53-71. doi: 10.1016/j.aeolia.2013.09.002

Marcolli, C., S. Gedamke, T. Peter, B. Zobrist (2007): Efficiency of immersion mode ice nucleation on surrogates of mineral dust. Atmos. Chem. Phys. 7, 5081-5091. doi: 10.5194/acp-7-5081-2007

Matsuki, A., Y. Iwasaka, G.-Y. Shi, H.-B. Chen, K. Osada, D. Zhang, M. Kido, Y. Inomata, Y.-S. Kim, D. Trochkine (2005): Heterogeneous sulfate formation on dust surface and its dependence on mineralogy: balloon-borne observations from balloon-borne measurements in the surface atmosphere of Beijing, China. Water Air Soil Pollut. 5(3), 101-132. doi: 10.1007/s11267-005-0730-3

May, K. R., R. Clifford (1967): The impaction of aerosol particles on cylinders, spheres, ribbons and discs. Ann. Occup. Hyg. 10, 83-95.

McInnes, L. M., D. S. Covert, P. K. Quinn, M. S. Germani (1994): Measurements of chloride depletion and sulfur enrichment in individual sea-salt particles collected from the remote marine boundary layer. J. Geophys. Res. 99(D4), 8257-8268. doi: 10.1029/93jd03453

Moreno, T., X. Querol, S. Castillo, A. Alastuey, E. Cuevas, L. Herrmann, M. Mounkaila, J. Elvira, W. Gibbons (2006): Geochemical variations in aeolian mineral particles from the Sahara–Sahel Dust Corridor. Chemosphere 65, 261-270. doi: 10.1016/j.chemosphere.2006.02.052

Munson, B. R., T. H. Okiishi, W. W. Huebsch, A. P. Rothmayer (2013): Fundamentals of Fluid Mechanics, 7th Edition, John Wiley & Sons.

Niemand, M., O. Möhler, B. Vogel, H. Vogel, C. Hoose, P. Connolly, H. Klein, H. Bingemer, P. DeMott, J. Skrotzki, T. Leisner (2012): A Particle-Surface-Area-Based Parameterization of Immersion Freezing on Desert Dust Particles. J. Atmos. Sci. 69(10), 3077-3092. doi: 10.1175/jas-d-11-0249.1

Niimura, N., K. Okada, X.-B. Fan, K. Kai, K. Arao, G.-Y. Shi, S. Takahashi (1998): Formation of Asian Dust-Storm Particles Mixed Internally with Sea Salt in the Atmosphere. J. Meteor. Soc. Japan 76(2), 275-288.

NOAA-ARL. (2017). "GDAS half-degree archive." 2016, from ftp://arlftp.arlhq.noaa.gov/pub/archives/gdas0p5, accessed 2016 and 2017.



Noll, K. E. (1970): A rotary inertial impactor for sampling giant particles in the atmosphere. Atmos. Environ. 4, 9-19. doi: 10.1016/0004-6981(70)90050-8

Noll, K. E., K. Y. P. Fang (1989): Development of a dry deposition model for atmospheric coarse particles. Atmos. Environ. 23(3), 585-594. doi: 10.1016/0004-6981(89)90007-3

Noll, K. E., M. M. Jackson, A. K. Oskouie (2001): Development of an Atmospheric Particle Dry Deposition Model. Aerosol Sci. Technol. 35(2), 627-636. doi: 10.1080/02786820119835

Noll, K. E., A. Pontius, R. Frey, M. Gould (1985): Comparison of atmospheric coarse particles at an urban and non-urban site. Atmos. Environ. 19(11), 1931-1943.

Nowottnick, E., P. Colarco, A. da Silva, D. Hlavka, M. McGill (2011): The fate of saharan dust across the atlantic and implications for a central american dust barrier. Atmos. Chem. Phys. 11(16), 8415-8431. doi: 10.5194/acp-11-8415-2011

Ott, D. K., W. Cyrs, T. M. Peters (2008a): Passive measurement of coarse particulate matter, $PM_{10-2.5}$. J. Aerosol Sci. 39(2), 156-167. doi: 10.1016/j.jaerosci.2007.11.002

Ott, D. K., T. Peters (2008b): A Shelter to Protect a Passive Sampler for Coarse Particulate Matter, $PM_{10-2.5}$. Aerosol Sci. Technol. 42, 299-309. doi: 10.1080/02786820802054236

Paik, S., J. H. Vincent (2002): Aspiration efficiency for thin-walled nozzles facing the wind and for very high velocity ratios. J. Aerosol Sci. 33(5), 705-720. doi: 10.1016/S0021-8502(01)00208-7

Pattengale, N. D., M. Alipour, O. R. P. Bininda-Emonds, B. M. E. Moret, A. Stamatakis (2010): How Many Bootstrap Replicates Are Necessary? J. Comput. Biol. 17(3), 337-354. doi: 10.1089/cmb.2009.0179

Petroff, A., L. Zhang (2010): Development and validation of a size-resolved particle dry deposition scheme for application in aerosol transport models. Geosci. Model Dev. 3(2), 753-769. doi: 10.5194/gmd-3-753-2010

Petters, M. D., S. M. Kreidenweis (2007): A single parameter representation of hygroscopic growth and cloud condensation nucleus activity. Atmos. Chem. Phys. 7(8), 1961-1971. doi: 10.5194/acp-7-1961-2007

Pinsky, M., A. Khain, M. Shapiro (1999): Collisions of Small Drops in a Turbulent Flow. Part I: Collision Efficiency. Problem Formulation and Preliminary Results. J. Atmos. Sci. 56(15), 2585-2600. doi: 10.1175/1520-0469(1999)056<2585:cosdia>2.0.co;2

Piskunov, V. N. (2009): Parameterization of aerosol dry deposition velocities onto smooth and rough surfaces. J. Aerosol Sci. 40(8), 664-679. doi: 10.1016/j.jaerosci.2009.04.006

Prospero, J. M., R. Arimoto (2009). Atmospheric Transport and Deposition of Particulate Material to the Oceans A2 - Steele, John H. Encyclopedia of Ocean Sciences (Second Edition). Oxford, Academic Press: 248-257.

Prospero, J. M., F.-X. Collard, J. Molinié, A. Jeannot (2014): Characterizing the annual cycle of African dust transport to the Caribbean Basin and South America and its impact on the environment and air quality. Global Biogeochem. Cy. 28(7), 2013GB004802. doi: 10.1002/2013gb004802

Rasband, W. S. (2015). "ImageJ." 1.47c. 2015.

Reid, E. A., J. S. Reid, M. M. Meier, M. R. Dunlap, S. S. Cliff, A. Broumas, K. Perry, H. Maring (2003): Characterization of African dust transported to Puerto Rico by individual particle and size segregated bulk analysis. J. Geophys. Res. 108(D19), 8591. doi: 10.1029/2002JD002935

Ro, C.-U., J. Osán, I. Szalóki, J. de Hoog, A. Worobiec, R. Van Grieken (2003): A Monte Carlo Program for Quantitative Electron-Induced X-ray Analysis of Individual Particles. Anal. Chem 75, 851-859.

Salter, M. E., E. Hamacher-Barth, C. Leck, J. Werner, C. M. Johnson, I. Riipinen, E. D. Nilsson, P. Zieger (2016): Calcium enrichment in sea spray aerosol particles. Geophysical Research Letters 43(15), 8277-8285. doi: 10.1002/2016gl070275

Savoie, D. L., J. M. Prospero, S. J. Oltmans, W. C. Graustein, K. K. Turekian, J. T. Merrill, H. Levy (1992): Sources of nitrate and ozone in the marine boundary layer of the tropical north Atlantic. J. Geophys. Res. 97(D11), 11575-11589. doi: 10.1029/92jd00894


Scheuvens, D., K. Kandler, M. Küpper, K. Lieke, S. Zorn, M. Ebert, L. Schütz, S. Weinbruch (2011): Indiviual-particle analysis of airborne dust samples collected over Morocco in 2006 during SAMUM 1. Tellus 63B, 512-530. doi: 10.1111/j.1600-0889.2011.00554.x

Schladitz, A., T. Müller, A. Nowak, K. Kandler, K. Lieke, A. Massling, A. Wiedensohler (2011): In situ aerosol characterization at Cape Verde Part 1: Particle number size distributions, hygroscopic growth and state of mixing of the marine and Saharan dust aerosol. Tellus 63B, 531-548. doi: 10.1111/j.1600-0889.2011.00569.x

Schütz, L. (1980): Long range transport of desert dust with special emphasis on the Sahara. Ann. NY Acad. Sci. 338(1), 515-532. doi: 10.1111/j.1749-6632.1980.tb17144.x

Sedwick, P. N., E. R. Sholkovitz, T. M. Church (2007): Impact of anthropogenic combustion emissions on the fractional solubility of aerosol iron: Evidence from the Sargasso Sea. Geochem. Geophy. Geosy. 8(10). doi: doi:10.1029/2007GC001586

Sehmel, G. A. (1973). Particle Deposition and Diffusivities Along Smooth Surfaces. Pollution. E. S. Barrekette, Springer. **2:** 564-571.

Shi, Z., M. D. Krom, S. Bonneville, A. R. Baker, C. Bristow, N. Drake, G. Mann, K. Carslaw, J. B. McQuaid, T. Jickells, L. G. Benning (2011a): Influence of chemical weathering and aging of iron oxides on the potential iron solubility of Saharan dust during simulated atmospheric processing. Global Biogeochem. Cy. 25, GB2010. doi: 10.1029/2010GB003837

Shi, Z., M. D. Krom, T. D. Jickells, S. Bonneville, K. S. Carslaw, N. Mihalopoulos, A. R. Baker, L. G. Benning (2012): Impacts on iron solubility in the mineral dust by processes in the source region and the atmosphere: A review. Aeolian Res. 5, 21-42. doi: 10.1016/j.aeolia.2012.03.001

Shi, Z. B., M. T. Woodhouse, K. S. Carslaw, M. D. Krom, G. W. Mann, A. R. Baker, I. Savov, G. R. Fones, B. Brooks, N. Drake, T. D. Jickells, L. G. Benning (2011b): Minor effect of physical size sorting on iron solubility of transported mineral dust. Atmos. Chem. Phys. 11(16), 8459-8469. doi: 10.5194/acp-11-8459-2011

Sholkovitz, E. R., P. N. Sedwick, T. M. Church, A. R. Baker, C. F. Powell (2012): Fractional solubility of aerosol iron: Synthesis of a global-scale data set. Geochim. Cosmochim. Acta 89, 173-189. doi: 10.1016/j.gca.2012.04.022

Slinn, S. A., W. G. N. Slinn (1980): Predictions for particles deposition on natural waters. Atmos. Environ. 14, 1013-1016. doi: 10.1016/0004-6981(80)90032-3

Stein, A. F., R. R. Draxler, G. D. Rolph, B. J. B. Stunder, M. D. Cohen, F. Ngan (2015): NOAA's HYSPLIT Atmospheric Transport and Dispersion Modeling System. Bull. Am. Met. Soc. 96(12), 2059-2077. doi: 10.1175/bams-d-14-00110.1

Stevens, B., D. Farrell, L. Hirsch, F. Jansen, I. Nuijens, I. Serikov, B. Brügmann, M. Forde, H. Linne, K. Lonitz, J. M. Prospero (2016): The Barbados Cloud Observatory: Anchoring Investigations of Clouds and Circulation on the Edge of the ITCZ. Bull. Am. Met. Soc. 97(5), 787-801. doi: 10.1175/bams-d-14-00247.1

Sullivan, R. C., S. A. Guazzotti, D. A. Sodeman, K. A. Prather (2007a): Direct observations of the atmospheric processing of Asian mineral dust. Atmos. Chem. Phys. 7, 1213-1236. doi: 10.5194/acp-7-1213-2007

Sullivan, R. C., S. A. Guazzotti, D. A. Sodeman, Y. Tang, G. R. Carmichael, K. A. Prather (2007b): Mineral dust is a sink for chlorine in the marine boundary layer. Atmos. Environ. 41, 7166-7179. doi: 10.1016/j.atmosenv.2007.05.047

Sundaram, S., L. R. Collins (1997): Collision statistics in an isotropic particle-laden turbulent suspension. Part 1. Direct numerical simulations. J. Fluid Mech. 335, 75-109. doi: 10.1017/s0022112096004454

Tang, M., D. J. Cziczo, V. H. Grassian (2016): Interactions of Water with Mineral Dust Aerosol: Water Adsorption, Hygroscopicity, Cloud Condensation, and Ice Nucleation. Chem. Rev. 116(7), 4205-4259. doi: 10.1021/acs.chemrev.5b00529



Trapp, J. M., F. J. Millero, J. M. Prospero (2010): Temporal variability of the elemental composition of African dust measured in trade wind aerosols at Barbados and Miami. Mar. Chem. 120, 71-82. doi: 10.1016/j.marchem.2008.10.004

Trincavelli, J., S. Limandri, R. Bonetto (2014): Standardless quantification methods in electron probe microanalysis. Spectrochim. Acta B 101, 76-85. doi: 10.1016/j.sab.2014.07.016

Ullerstam, M., R. Vogt, S. Langer, E. Ljungström (2002): The kinetics and mechanism of $SO_2$ oxidation by $O_3$ on mineral dust. Phys. Chem. Chem. Phys. 4, 4694-4699. doi: 10.1039/B203529B

Usher, C. R., H. Al-Hosney, S. Carlos-Cuellar, V. H. Grassian (2002): A laboratory study of the heterogeneous uptake and oxidation of sulfur dioxide on mineral dust particles. J. Geophys. Res. 107(D23), 4713. doi: 10.1029/2002JD002051

van der Does, M., L. F. Korte, C. I. Munday, G. J. A. Brummer, J. B. W. Stuut (2016): Particle size traces modern Saharan dust transport and deposition across the equatorial North Atlantic. Atmos. Chem. Phys. 16(21), 13697-13710. doi: 10.5194/acp-16-13697-2016

Wagner, J., D. Leith (2001): Passive Aerosol Sampler. Part I: Principle of Operation. Aerosol Sci. Technol. 34(186-192). doi: 10.1080/027868201300034808

Wang, L.-P., O. Ayala, S. E. Kasprzak, W. W. Grabowski (2005): Theoretical Formulation of Collision Rate and Collision Efficiency of Hydrodynamically Interacting Cloud Droplets in Turbulent Atmosphere. J. Atmos. Sci. 62(7), 2433-2450. doi: 10.1175/jas3492.1

Warneck, P., J. Williams (2012): The Atmospheric Chemist's Companion. Numerical Data for Use in the Atmospheric Sciences, Springer Netherlands.

Weinbruch, S., M. Wentzel, M. Kluckner, P. Hoffmann, H. M. Ortner (1997): Characterization of Individual Atmospheric Particles by Element Mapping in Electron Probe Microanalysis. Mikrochim. Acta 125, 137-141. doi: 10.1007/BF01246176

Weinzierl, B., A. Ansmann, J. M. Prospero, D. Althausen, N. Benker, F. Chouza, M. Dollner, D. Farrell, W. K. Fomba, V. Freudenthaler, J. Gasteiger, S. Groß, M. Haarig, B. Heinold, K. Kandler, T. B. Kristensen, O. L. Mayol-Bracero, T. Müller, O. Reitebuch, D. Sauer, A. Schäfler, K. Schepanski, A. Spanu, I. Tegen, C. Toledano, A. Walser (2017): The Saharan Aerosol Long-Range Transport and Aerosol–Cloud-Interaction Experiment: Overview and Selected Highlights. Bulletin of the American Meteorological Society 98(7), 1427-1451. doi: 10.1175/bams-d-15-00142.1

Weinzierl, B., D. Sauer, M. Esselborn, A. Petzold, A. Veira, M. Rose, S. Mund, M. Wirth, A. Ansmann, M. Tesche, S. Gross, V. Freudenthaler (2011): Microphysical and optical properties of dust and tropical biomass burning aerosol layers in the Cape Verde region—an overview of the airborne in situ and lidar measurements during SAMUM-2. Tellus B 63(4), 589-618. doi: 10.1111/j.1600-0889.2011.00566.x

Wen, X., D. B. Ingham (2000): Aspiration efficiency of a thin-walled cylindrical aerosol sampler at yaw orientations with repsect to the wind. J. Aerosol Sci. 31(11), 1355-1365. doi: 10.1016/S0021-8502(00)00036-7

Wood, N. B. (1981): A simple method for the calculation of turbulent deposition to smooth and rough surfaces. J. Aerosol Sci. 12(3), 275-290. doi: 10.1016/0021-8502(81)90127-0

Worringen, A., K. Kandler, N. Benker, T. Dirsch, S. Mertes, L. Schenk, U. Kästner, F. Frank, B. Nillius, U. Bundke, D. Rose, J. Curtius, P. Kupiszewski, E. Weingartner, P. Vochezer, J. Schneider, S. Schmidt, S. Weinbruch, M. Ebert (2015): Single-particle characterization of ice-nucleating particles and ice particle residuals sampled by three different techniques. Atmos. Chem. Phys. 15(8), 4161-4178. doi: 10.5194/acp-15-4161-2015

Xu, X., U. S. Akhtar (2010): Identification of potential regional sources of atmospheric total gaseous mercury in Windsor, Ontario, Canada using hybrid receptor modeling. Atmos. Chem. Phys. 10(15), 7073-7083. doi: 10.5194/acp-10-7073-2010

Zamora, L. M., J. M. Prospero, D. A. Hansell (2011): Organic nitrogen in aerosols and precipitation at Barbados and Miami: Implications regarding sources, transport and deposition to the western subtropical North Atlantic. J. Geophys. Res. 116(D20). doi: doi:10.1029/2011JD015660



Zhang, D. (2008): Effect of sea salt on dust settling to the ocean. Tellus 60B, 641-646. doi:
10.1111/j.1600-0889.2008.00358.x
Zhang, D., Y. Iwasaka (1999): Nitrate and sulfate in individual Asian dust-storm particles in Beijing,
China in spring of 1995 and 1996. Atmos. Environ. 33, 3213-3223. doi: 10.1016/S1352-
2310(99)00116-8
Zhang, D., Y. Iwasaka (2004): Size change of Asian dust particles caused by sea salt interaction:
Measurements in southwestern Japan. Geophys. Res. Lett. 31, L15102. doi:
10.1029/2004GL020087








**Table 1: Relationships between area coverage of the simulated 5 mm x 5 mm analysis field, particle numbers, particle masses and uncertainties. Upper part: CV-ground size distribution, lower part: CV-air size distribution. A bulk density of 2500 kg/m³ was assumed for the mass estimation from particle volume. Abbreviations: SP coverage = ratio of the sum of single particle cross sections to the analysis field; apparent coverage = fraction of area covered by the particles after deposition; $N_{>1}$ = Number of particles larger than 1 μm diameter; $PM_{>1}$ = Total mass of particles larger than 1 μm diameter (approx. 99.99 % of total mass); $PM_{1-32}$ = total mass of particles between 1 μm and 32 μm diameter (approx. 50 % of total mass for source-near size distribution, 67 % for aged one). Relative uncertainty is given as the ratio of the upper and lower bounds of the central 95 % quantile to the median of 1000 (200 for SP coverage >= 0.1) repetitions of deposition simulation.**

| SP coverage | Apparent coverage | $N_{>1}$ | $PM_{>1}$, μg | relative uncertainty | $PM_{1-32}$, μg | relative uncertainty |
|---|---|---|---|---|---|---|
| 0.001 | 0.001 | 353 | 0.8 | 0.38 - 4.16 | 0.5 | 0.56 - 1.38 |
| 0.003 | 0.002 | 865 | 2.7 | 0.37 - 4.68 | 1.3 | 0.69 - 1.24 |
| 0.005 | 0.005 | 1699 | 4.6 | 0.53 - 3.60 | 2.5 | 0.78 - 1.18 |
| 0.006 | 0.006 | 2032 | 6.1 | 0.49 - 2.80 | 3.0 | 0.81 - 1.17 |
| 0.007 | 0.007 | 2361 | 6.9 | 0.54 - 2.76 | 3.5 | 0.82 - 1.16 |
| 0.008 | 0.008 | 2692 | 7.6 | 0.56 - 2.59 | 4.0 | 0.83 - 1.14 |
| 0.009 | 0.009 | 3016 | 8.3 | 0.58 - 2.57 | 4.5 | 0.84 - 1.14 |
| 0.010 | 0.010 | 3344 | 10 | 0.54 - 2.21 | 5.0 | 0.85 - 1.13 |
| 0.011 | 0.011 | 3669 | 11 | 0.57 - 2.26 | 5.5 | 0.85 - 1.13 |
| 0.012 | 0.012 | 3988 | 11 | 0.63 - 2.18 | 5.9 | 0.86 - 1.12 |
| 0.013 | 0.013 | 4313 | 13 | 0.64 - 2.03 | 6.4 | 0.87 - 1.12 |
| 0.015 | 0.015 | 4951 | 14 | 0.67 - 1.95 | 7.4 | 0.88 - 1.11 |
| 0.020 | 0.020 | 6520 | 20 | 0.69 - 2.09 | 9.8 | 0.89 - 1.10 |
| 0.025 | 0.025 | 8047 | 24 | 0.74 - 1.94 | 12 | 0.90 - 1.09 |
| 0.035 | 0.034 | 10998 | 34 | 0.77 - 1.71 | 17 | 0.92 - 1.07 |
| 0.050 | 0.048 | 15146 | 56 | 0.69 - 1.48 | 24 | 0.93 - 1.06 |
| 0.075 | 0.071 | 21379 | 81 | 0.75 - 1.45 | 36 | 0.94 - 1.05 |
| 0.100 | 0.093 | 26824 | 106 | 0.79 - 1.35 | 47 | 0.95 - 1.05 |
| 0.200 | 0.172 | 34099 | 218 | 0.80 - 1.53 | 89 | 0.97 - 1.04 |
| 0.001 | 0.001 | 1031 | 0.7 | 0.44 - 3.05 | 0.5 | 0.66 - 1.33 |
| 0.005 | 0.005 | 5056 | 3.6 | 0.73 - 1.84 | 2.4 | 0.84 - 1.15 |
| 0.010 | 0.010 | 9990 | 8.1 | 0.71 - 1.42 | 4.8 | 0.89 - 1.11 |
| 0.025 | 0.025 | 24102 | 19.8 | 0.79 - 1.27 | 11.9 | 0.93 - 1.07 |
| 0.050 | 0.048 | 45618 | 39.4 | 0.82 - 1.38 | 23.4 | 0.95 - 1.05 |
| 0.075 | 0.071 | 64665 | 59.1 | 0.85 - 1.29 | 34.6 | 0.96 - 1.04 |
| 0.100 | 0.093 | 81568 | 78.5 | 0.89 - 1.23 | 45.4 | 0.97 - 1.04 |
| 0.150 | 0.134 | 109224 | 118 | 0.90 - 1.17 | 65.7 | 0.97 - 1.03 |
| 0.200 | 0.173 | 129769 | 158 | 0.91 - 1.13 | 84.2 | 0.97 - 1.02 |






**Table 2: Average dust mass concentrations estimated from deposited particle mass applying various deposition models.**
**Lower and upper refer to different dust fraction estimates (see Eq. (6) and (12)).**

| Model | lower estimate, µg/m³ | upper estimate, µg/m³ |
|---|---|---|
| **Stokes settling** | 149 | 195 |
| **Noll et al. (2001)** | 0.28 | 0.32 |
| **Noll et al. (1989)** | 67 | 96 |
| **Aluko et al. (2006)** | 58 | 85 |
| **Piskunov (2009)** | 32 | 47 |
| **Wagner et al. (2001)** | 81 | 115 |
| **High-volume sampler** | 26 | |









**Fig. 1:** Comparison of the Na/Cl ratio of sodium chloride powder as function of particle size, corrected by the methods
Cliff-Lorrimer, ZAF and interpolated. Measurements were performed at 20 kV acceleration voltage. The sodium chloride
nominal ratio is shown as orange line. The linear regression of the interpolated correction is shown as black striped line.

**Fig. 2:** Calculated ion balance for all beam interaction volumes containing particles dominated by Na and Cl. Particles
were collected by the DPDS. The axes are scaled in arbitrary units of percent × unit charges. Smaller particles yield
smaller values as they only fill a fraction of the beam interaction volume. Particle size is color-coded; note that all
particles between 0.6 μm and 1 μm in size are shown as blue, and between 10 μm and 25 μm as red. The black diagonal
lines show the 10 % deviation cone.

**Fig. 3:** Mean element index only using Na, Mg, S, Cl, and Ca for normalization, and according standard deviation (1 σ) for
NaCl-dominated particles from a typical atmospheric sample as function of particle size. Note that relative standard
deviation for Ca is not shown due to frequent values below the detection limit.

**Fig. 4:** Comparison of the relative two-sided 95 % confidence interval limits for bootstrap and Poisson approaches. Values
shown are the confidence interval limits for the total deposited particle volume divided by this volume. Data basis are
the deposition samples at Ragged Point of 2013. Left: for all particles; right: for particles between 1 μm and 20 μm
diameter. The color shows the fraction of the total volume present in the single largest particle. Note the different scales
between the graphs.

**Fig. 5:** Deposition velocity to a smooth surface calculated by different deposition models for the samples of 2013, taking
into account the ambient thermodynamic conditions and the particle composition. a: Stokes settling; b: Noll et al. (2001);
c: Noll et al. (1989); d: Aluko et al. (2006); e: Piskunov (2009) ; f: Wagner et al. (2001)

**Fig. 6:** Number (left column) and volume size distributions (right column) of deposition rates as function of projected
area diameter modeled for Cape Verde aerosol as derived from a 5 mm x 5 mm analysis field. The upper row is based on
CV-ground, the lower on CV-air size distributions. The grey curve shows the original size distribution of deposited
particles, the colored points with whiskers give median and central 95 % quantile of 1000 repetitions (200 for 0.093 and
0.172/0.173) for distributions calculated from samples with different area fractions covered by particles.

**Fig. 7:** Number (left column) and volume size distributions (right column) of deposited particles measured at Ragged
Point and extrapolated change as function of particle projected area diameter and area coverage fraction, simulating a
longer exposure time. Different colors show different factors of exposure increase (5x, 10x, 15x, 20x). Resulting coverage
fractions are given in the figure keys.

**Fig. 8:** Particle volume per area calculated from single particle measurements as function of the fractional area coverage.
Blue symbols denote the unmodified samples, red symbols the simulation of higher coverage by factors of 2, 3, 5, 10, 15,
and 20. Error bars denote the two-sided 95 % confidence interval. The fit function shown as black dashed line is
calculated as $= exp(a\ ln(x) + b)$ ; $[y] = mm^3/m^2$; $a = 0.957 \pm 0.041$; $b = 3.57 \pm 0.06$; $x$ is the fractional area
coverage.






Fig. 9: Upper 95 % quantile of the fractions of internally mixed particles due to coincidental mixture on the substrate
(color scale) as function of the projected area diameter and substrate area coverage, for a two-component system.
Strong mixture refers to a minimum particle volume fraction of the other component of 20 %, detectable mixture refers
to 1 %. Ratios of the two components in the base aerosol are given as percentages above each plot.

1416

Fig. 10: Upper 95 % quantile of the fractions of internally mixed particles due to coincidental mixting on the substrate
(color scale), for a dust/sea-salt/sulfate system with measured composition and CV-ground size distribution. Strong
mixture refers to a minimum particle volume fraction of the other component of 20 %, detectable mixture refers to 1 %.
Mixing compounds are given on top of each graph. Sulfate/sea-salt and ternary mixtures practically do not form
coincidentally.

1422

Fig. 11: Average atmospheric mass size distribution densities derived from DPDS and FWI measurements. Left: period
from July 10 to 15, 2013; right: from June 14 to July 8, 2013. Different colors refer to different deposition velocity
estimates as shown in Fig. 5. Solid lines refer to total mass concentrations, dashed ones to the dust mass estimated from
the chemical composition (upper limit estimate). Error bars show the central 95 % confidence interval. Pink crosses show
a size distribution measured in the Saharan air layer on June 22, 2013 (Weinzierl et al. 2017). Note that for particles
smaller than 10 µm the FWI data may contain a considerable bias in calculation.

1429

Fig. 12: Dust mass concentration and flux density time series derived from DPDS compared to such obtained from high-
volume sampler (Kristensen et al. 2016). The darker brown bar shows the range from lower to upper estimate, the blue
triangles the lower and upper estimate of dust deposition flux density. The date refers to the year 2013.

1433

Fig. 13: Size dependence of the relative number abundance of major particle types as derived from single particle
analysis of deposited aerosol.

1436

Fig. 14: Left: box plot of daily mass deposition rate for total dust and dust-derived elements at Barbados for 2013 and
2016. Right: cumulative mass deposition flux as function of aerodynamic particle diameter for dust and dust-derived
elements. Note that for P and Ti in the latter plot two particles containing each more than 10 % of the total deposited
mass have been removed from the data set.

1441

Fig. 15: Potential source contribution functions (PSCF) of deposited material: dust (upper row), total soluble sulfate
(central row) and relative ion balance for sulfate particles (lower row) for 2013 and 2016 at Ragged Point. Note that for
dust, potential provenance is calculated for Saharan Air Layer transport only (i.e. trajectory arrival altitudes > 1500 m).

1445

Fig. 16: Iron availability (SIAI) and deposition potential source contribution functions (PSCF) for 2013 and 2016. Upper
row: geometric iron availability index, only silicate particles counted; trajectories arriving 1500–3000 m over Ragged
Point. Lower row: mass deposition rate of Fe-rich particles smaller than 4 µm in diameter; trajectories arriving lower
than 1500 m over Ragged Point.

1450

Fig. 17: Ca/Na atomic ratio as function of particle dry diameter for all sea-salt particles collected at Ragged Point in 2013.
Different samplers are shown by color: CI blue, DPDS red, and FWI brown





1453

**Fig. 18: Time-series of wind and particle number deposition rates for pure compounds and internally mixed particles for June/July 2013 and August 2016. Particle size ranges are given in the top left of each graph. The limit of detection for the number of internally mixed particles is shown as line in the according color. Where only the detection limit for silicate/sulfate mixtures is visible, both limits are identical.**

1458

**Fig. 19: Deposition velocities calculated with the Piskunov model for internal admixture of sea-salt (left) or sulfate (right) for the mixed particles observed at Ragged Point. Velocities are given for the unmixed dust core and internal mixtures at dry conditions, at ambient relative humidity, and at 90 % relative humidity. The lines show the according means. Note that variation in deposition velocity for the same dust core size arises from variation in wind speed and admixed fraction.**

1463

**Fig. 20: Effective deposition velocity for all dust-containing particles observed at Ragged Point. The blue curves take into account internal mixing and hygroscopic growth at ambient conditions, whereas the orange only regards the dry dust fraction of the particles. In addition, cumulative mass distribution is shown on the inverted right axis. Particle size is given as aerodynamic diameter for the dust fraction of a particle. For the ambient deposition velocity, the geometric mean for each size class in shown in conjunction with the 1 geometric standard deviation range.**

1469





Fig. 01

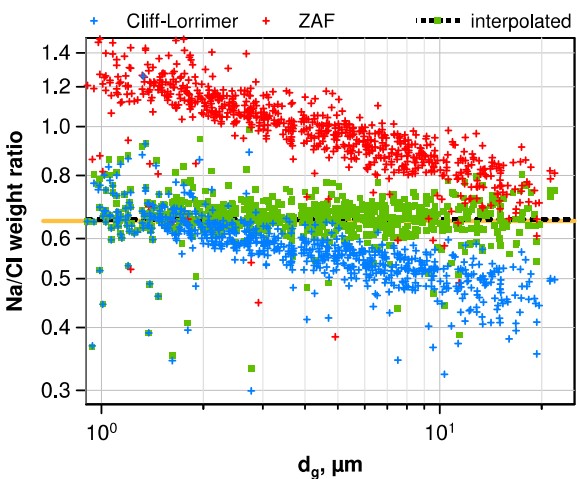





**Fig. 02**

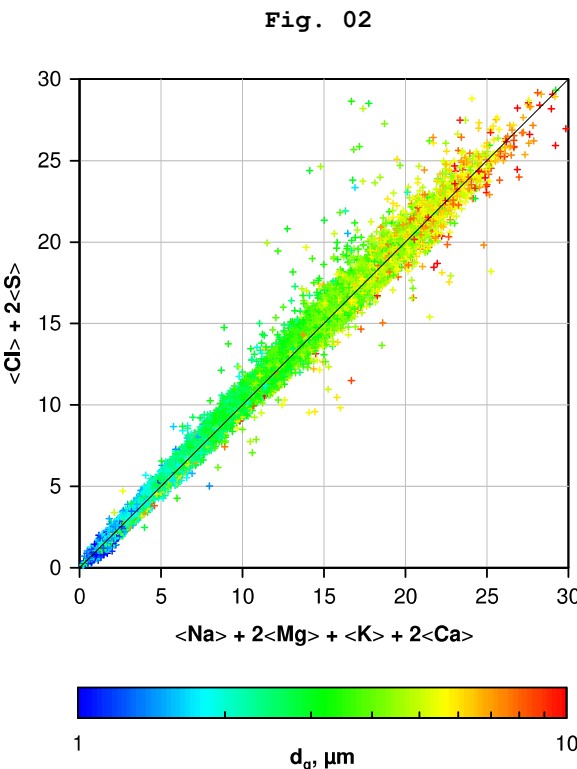



Fig. 03

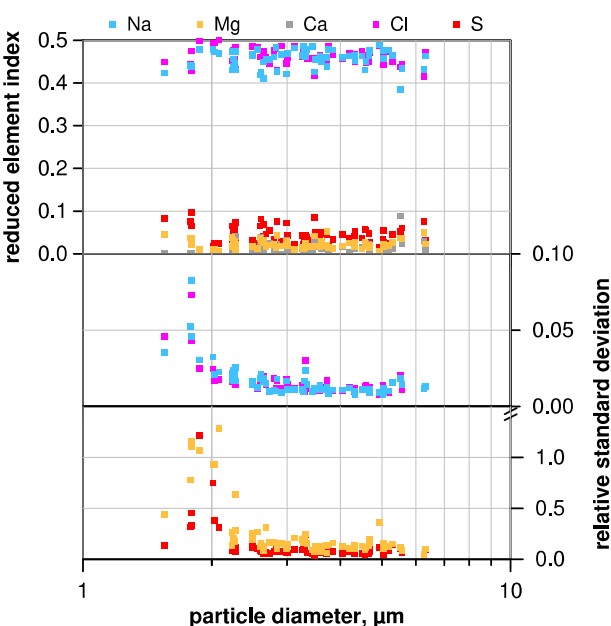

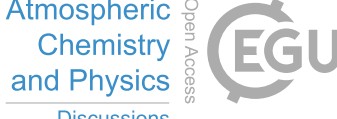

Fig. 04

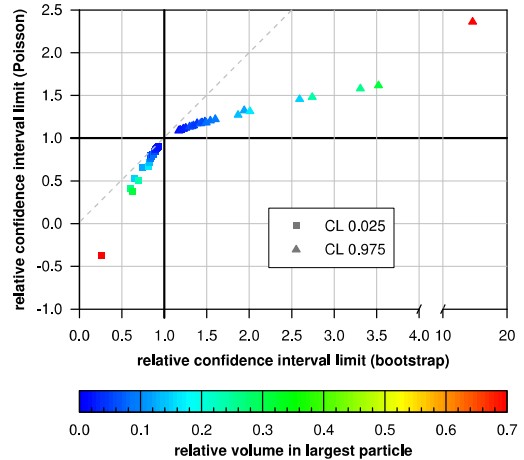
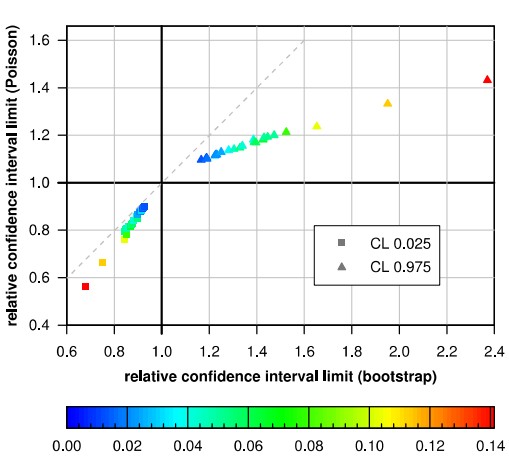





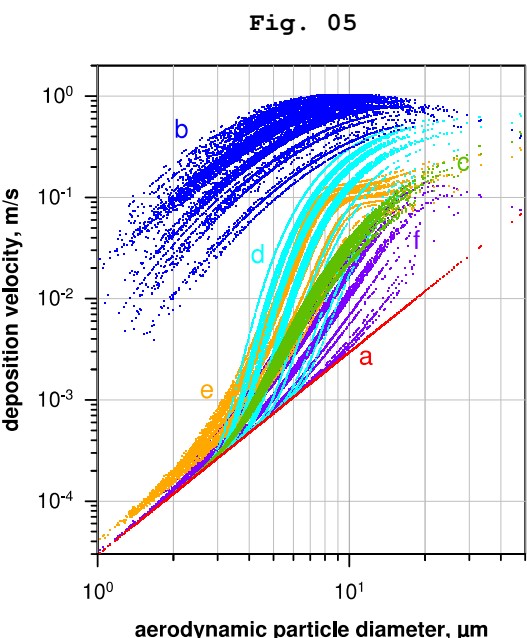

**Fig. 05**





Fig. 06

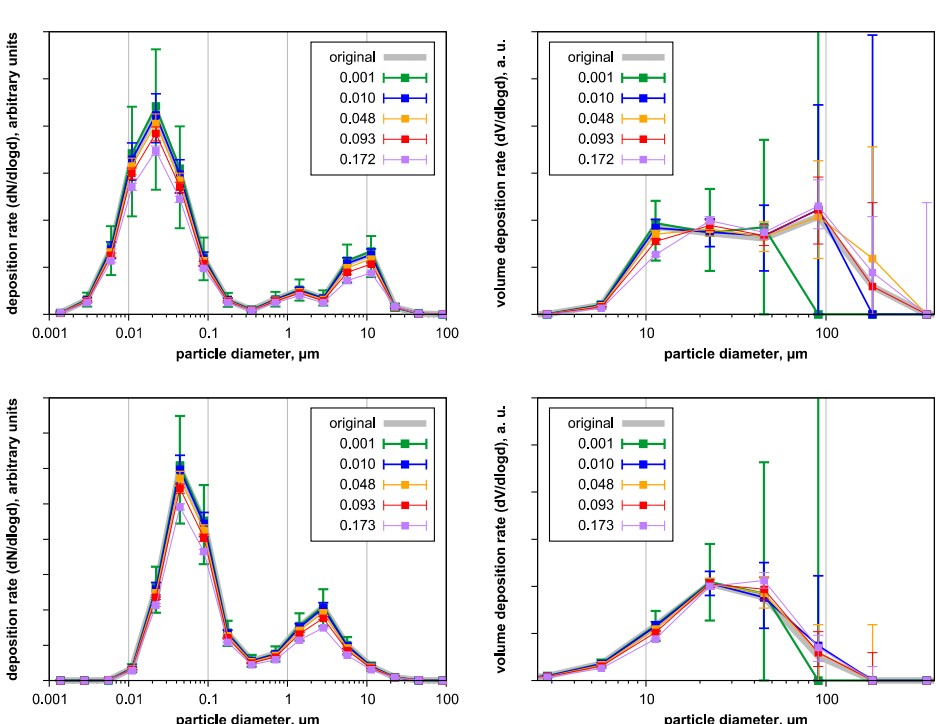



Fig. 07

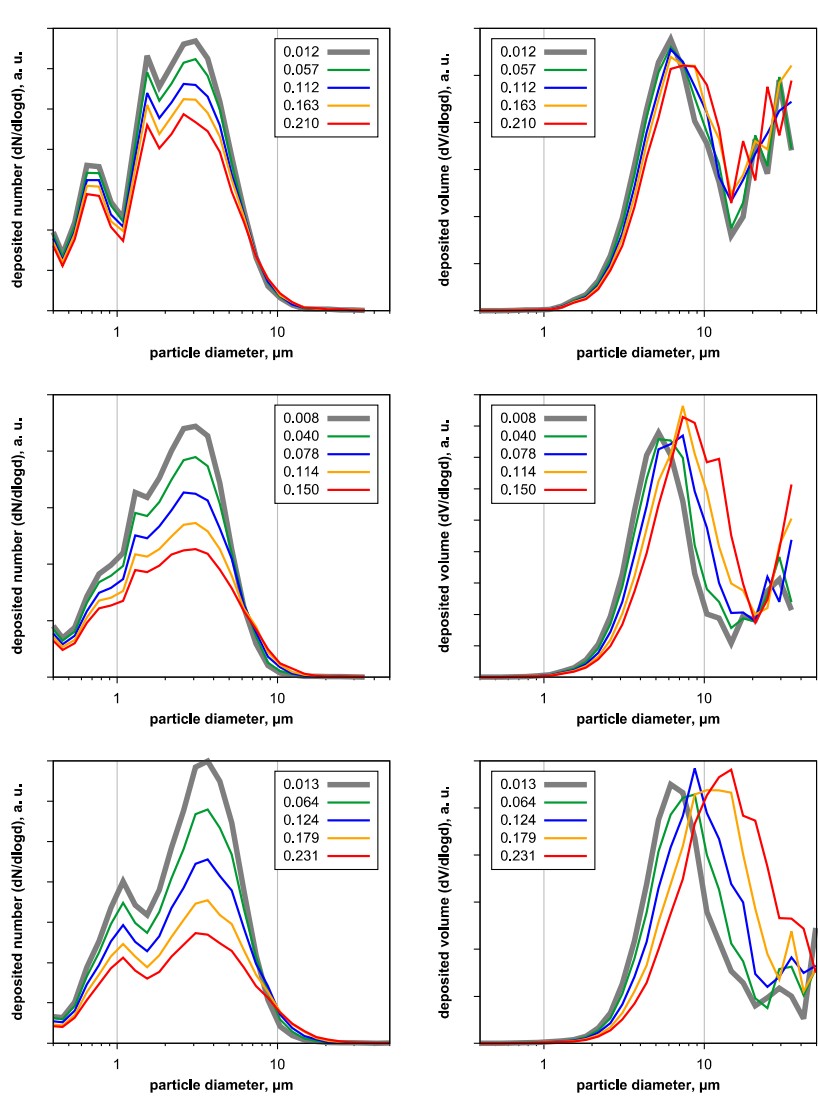





**Fig. 08**

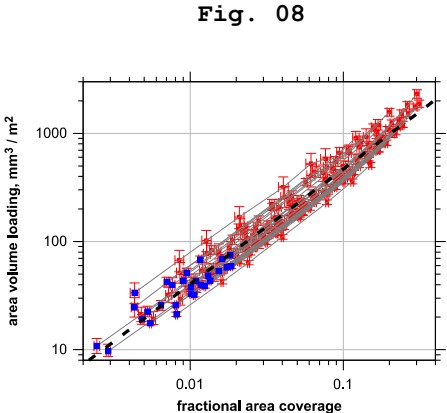



Fig. 09





Fig. 10

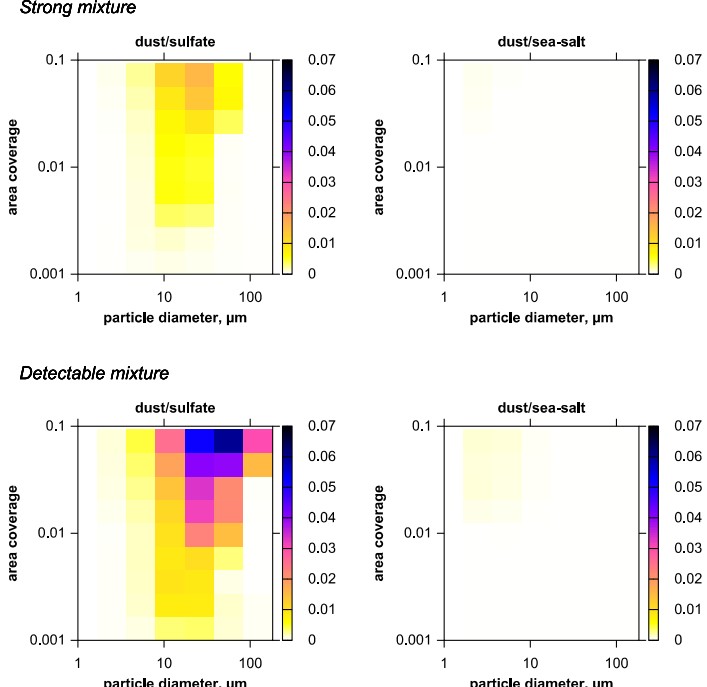



Fig. 11

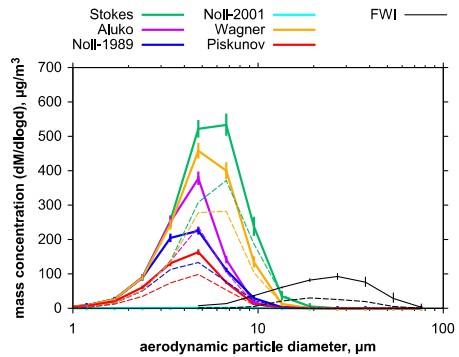
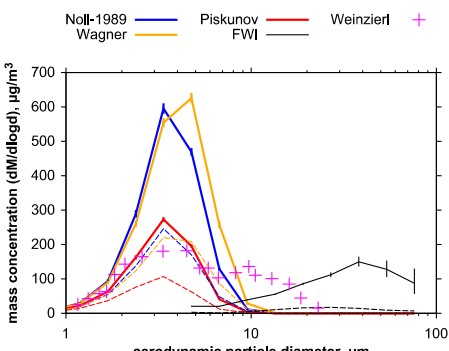



Fig. 12

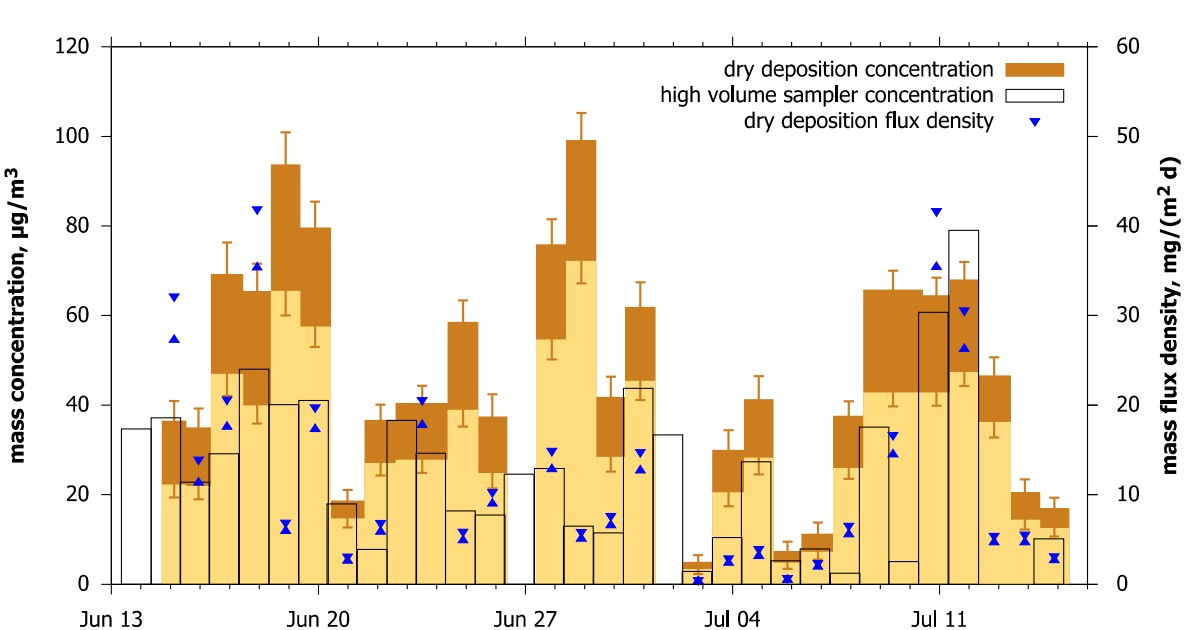




Fig. 13

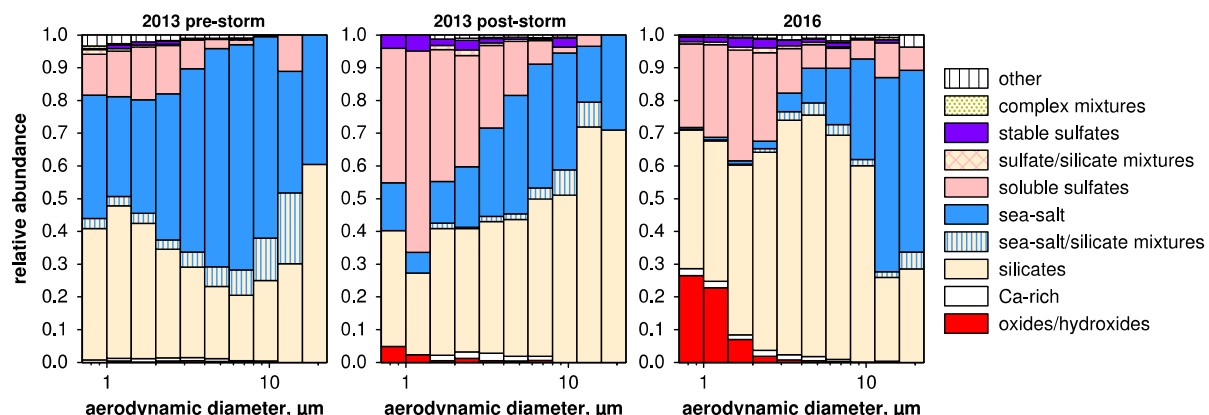





Fig. 14

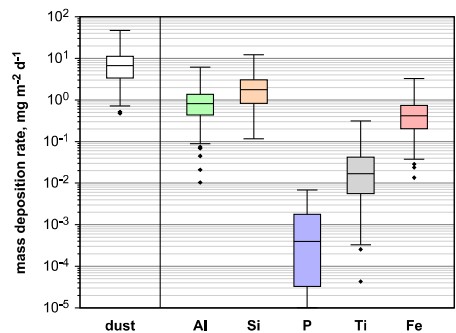
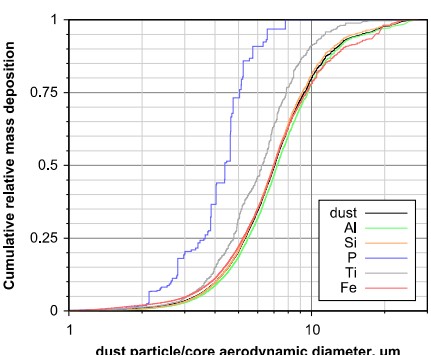





Fig. 15

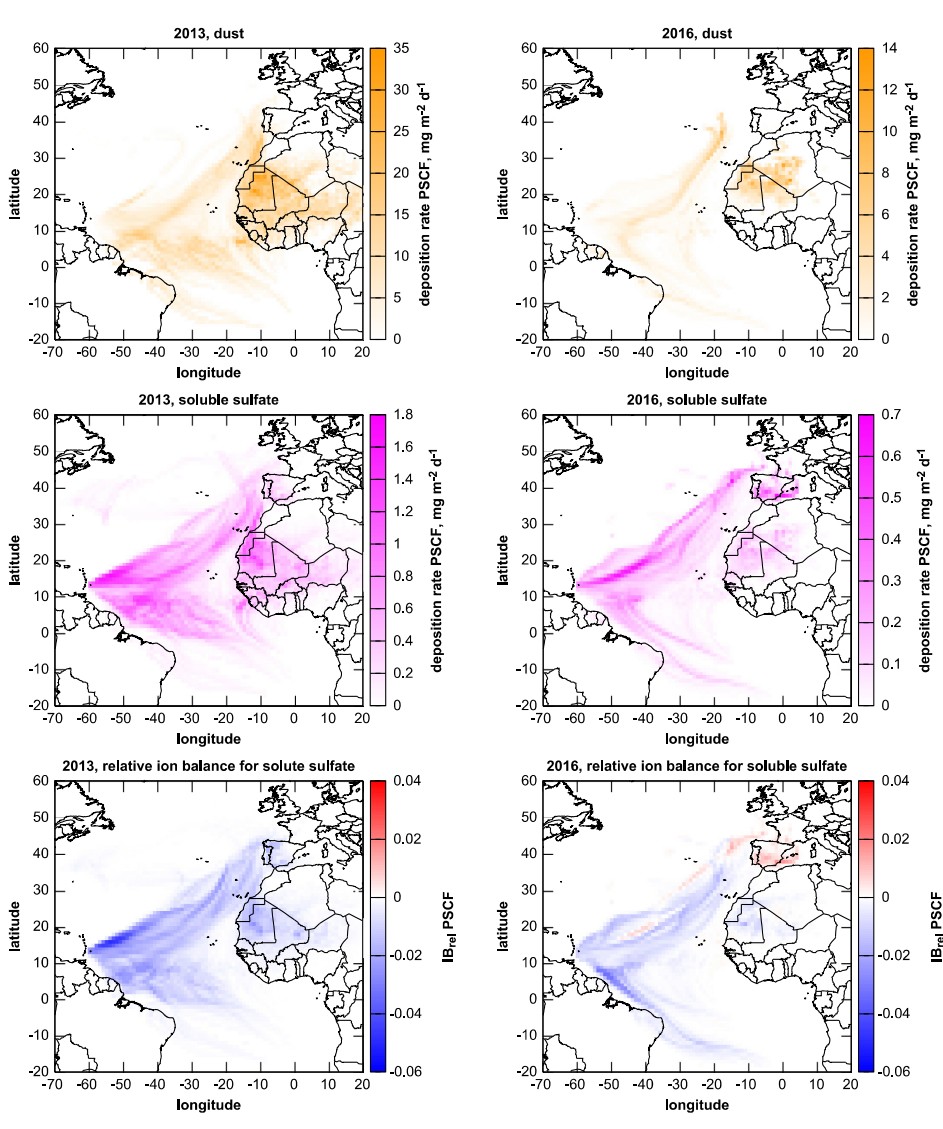





Fig. 16

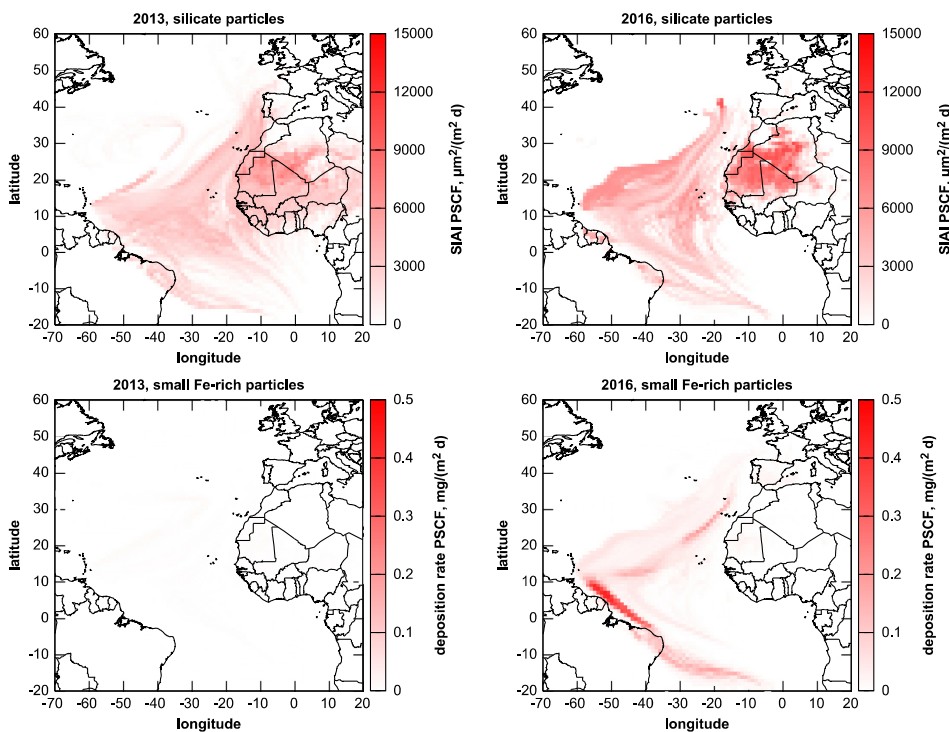

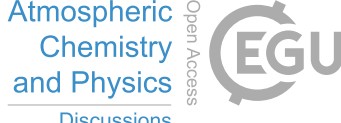



**Fig. 17**







Fig. 18







Fig. 19

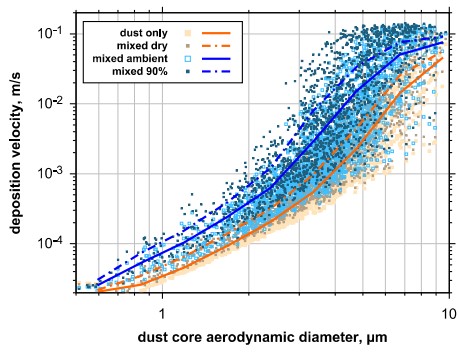 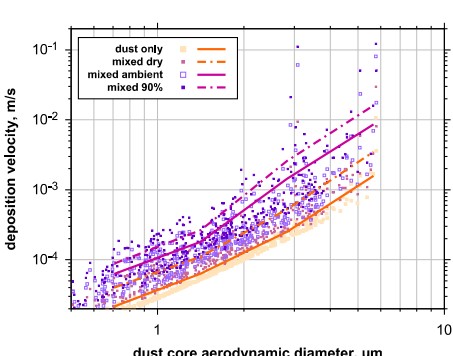



## Fig. 20

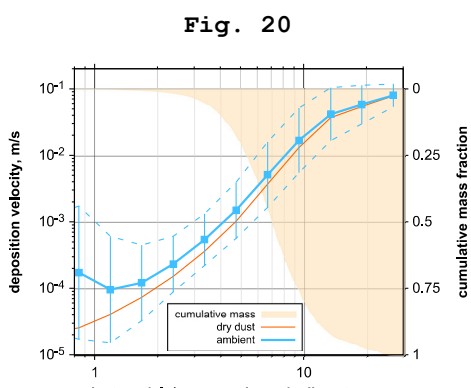