# Peer review of "Composition and mixing state of atmospheric aerosols determined by"

_Atmospheric Chemistry and Physics, 2018_

## Referee Comment (RC1) · Anonymous Referee #1 · 4 Jun 2018

**General comments** The authors have provided a very nice manuscript of measurements and associated modelling of aerosols in the Caribbean boundary layer which is suitable for publishing in ACP. The composition and mixing state of dust, sea salt and sulphate are analysed and they have been detailed in their efforts and provide a good analysis of uncertainty in the results. The general flow of the paper could be improved by condensing the methods and some small additions to the discussions. Of note I feel is the dry deposition discussion – a parameter which hinders aerosol modelling in

general and, as shown here, needs further work in constraining.

**Specific comments**
**Methods:**
The methods section is highly detailed, particularly in its use of equations. While it is commendable to be thorough in an analysis I feel the manuscript would greatly benefit from condensing the main methods to the core equations and text. Following each one through in detail to its final derivation is not necessary for the discussion and instead detracts the reader from getting to said results. The SI is the ideal place for much of this text, and some could even be removed totally.

For example,
Section 2.3.5: It is good to be thorough and test other statistics, but this is not necessary in the main text and somewhat disturbs the flow. It is sufficient to briefly state you performed the bootstrap analysis with 10,000 replications and move on.
Section 2.4.1: The Petters manuscript and hygroscopicity term is well known within the atmospheric community. Small statements about assumed values and uncertainties would be sufficient I feel.
**Results:**

Section 3.2.1: In Figure 5, model d (Aluko) looks like it would perhaps give closer agreement to results in Table 2 than it does. By comparing results of model e (Piskunov) and d (and others if appropriate, e.g. from the tuning exercise) could the Authors also comment on impacts that uncertainty in differences in the deposition velocity for fine (<2.5 um) or coarse (>2.5 um) mode aerosol has on the results? - which are significant for fine mode between models d and e, but more similar in coarse mode.
Section 3.3.2: This is a great section but would befit from a bit more explanation I

feel. How do the airmass providences link to the trajectories? Fig 15 looks similar to the trajectories, but Fig 16 (particularly the iron) has a very difference providence – how was this reached? This is important as the air mass trajectories suggest there is a difference in 2016 in that it has a much stronger European component to it than 2013. Which makes sense with the SO4 sources analysis. However, the southern African source the Authors state for the combustion iron is less obvious I feel from the trajectories themselves, although apparent in Fig 16. Furthermore, the total iron to dust correlation is unfortunately not shown but, as the authors point out, combustion iron has become a topic of much discussion recently and so it would therefore be good to see this result and then put in the context of whether a combustion iron source is visible in 2016 vs 2013. For a south American source can the Authors identify if this likely to be anthropogenic or biomass burning dominated?

Section 3.3.4: Small additional discussions about

1) Iron solubility: interstitial and cloud-borne changes with sulphate in particular.
2)Relative concentration of feldspar in the ice nucleation discussion.
3)Wet deposition as a loss process when activated to be CDNC. Would be interesting.

**Technical corrections**

**Figures**
Please add legends to all figures where missing and check the use of colours is appropriate (see below for some examples).
Fig 2: The use of a log scale and a continuous colour bar is not intuitive. Please change to a discrete colour bar (10 or 5 colours).
Fig 4: colours do not match numbers using this scale, e.g. green is 2.5-4.5, but blue is 1.2-1.8?

Fig 12: The empty box plot for high volume sampler concentration looks odd. I suggest replacing with a simple horizonal line or a shape (e.g. a star).

Fig 13: the x-axis is logscale, but the bars are a fixed width. This does not make sense. Either alter to scalar plot (preferred) or alter the width of the bars to match the scale.

Figs 15, 16: Again, the use of continuous colours would probably be better as discreet.

Fig 16: 2013 small Fe-rich particles looks like it missing the data?

Fig S4, S5. Increase legend size to a single bar and add numbers to it.

Fig S9 : see Fig 12 note above

Supplementary tables and figures can be grouped together (currently tables are interspersed throughout figures)

**Text**

*Italics are suggested additions to text.*

L35: Largest by mass only, not number.

L36-37: Expand this by a sentence, too brief.

L40: Define the Central American Dust Barrier causes.

L42: What processes are not fully understood? Use of 'these' is too vague here.

L44: Change to: '…by *physical and* chemical processing, …"

L46-47: This is quite obvious. Best to either expand or remove sentence.

L47-48: Link with expanded L36-37 as to why this happens.

L49: While could start a new paragraph.

L49-51: Brief summary of these studies/anything of importance to note?

L55 an on: change methodical to *methods*

L62: 'Offline' can mean many things. Define it here to avoid ambiguity.

L111: remove space before comma

L198: '…mentioned correction methods as*a* function …'

L199: 'a higher accuracy *in* …. can be achieved'

L246: '…not *the* focus ..'

L255: '...separately an important...'

L260: without seeing the dust:iron ratio it is not possible to say that Fe is safely assumed to be from dust and not combustion.

L540: 'conclude'

L722-723. Deposition velocities are described in section 2.4.2.

L792-793: 2016 listed twice. However, I can see no obvious difference in the air masses coming from South America in Fig.S6 anyhow?

L801-803: Nitrate as well as sulphate associated with dust sources is likely to be from Europe (e.g., anthropogenic in origin).
* * *

---

## Referee Comment (RC2) · Anonymous Referee #2 · 19 Jun 2018

General comments:

This manuscript presents new methods for accurately determining composition and mixing states of individual coarse aerosol particles by an automated SEM-EDX, and their application on the aged Saharan dust samples collected in the Caribbean boundary layer. Traditionally, individual particle analysis suffered from uncertainties related to poor counting statistics. There has been a significant gap between the ambient mass

size distribution and such qualitative information based on individual particle analysis. The authors however, succeeded to analyze large number of particles by the use of a novel automated SEM-EDX, and to provide a much more comprehensive and quantitative view on the mixing states of dust as a function of particle size. Sampling and data analysis were done very carefully by taking into account many potential sources of errors and uncertainties (e.g. quantification of elements, estimation of dust mass, sampling artifacts, counting statistics) and they are well defined and characterized in the manuscript. I therefore believe the manuscript deserves certain credit and can be a good contribution to ACP. However, such thorough and detailed verification of the methods in turn made the manuscript rather lengthy (especially the method section) and not easy for the readers to digest its major findings. It requires an extensive restructuring, and there are also some concerns on the assumptions made and interpretation of the results. The paper may be recommended for publication after these concerns are properly addressed. Specific comments are listed below:

Specific comments:

Lines 17-19: Please rephrase "Techniques were developed to conclude from collected aerosol on atmospheric concentrations and aerosol mixing state, and different models were compared."

Line 51: "Data basis is still limited" can be elaborated to make this study stand out more from previous studies.

Line 56: methodological?

Lines 68-623: While introduction section is made up of only 3 paragraphs, method section accounts for the majority of the manuscript and the overall structure appears rather unbalanced. Many details, equations and figures in the method section can be moved to supplementary information (SI). The explanation in the text can be significantly shortened by simply referring to corresponding sections in the SI.

[Figure]

Line 69: Sampling time and duration for three different samplers are not well explained. How do they coincide? I imagine deposition sampler requires longer time.

Line 147: As a result,

Lines 175-178: Validity of estimating volume equivalent diameter (in section 2.3) based only on 2-dimensional projected area and perimeter is questionable. It may work rather well for dry and solid particles such as pure silicates, but if particles were internally mixed with soluble materials and sampled at high relative humidity conditions (which might often be the case at Ragged Point), flattening of deliquesced particles on the substrate may become a source of significant sizing bias. Some people employ tilting of sample stages or apply shadowing to measure the height of the particles under SEM analysis.

Line 185: achieved instead of reached?

Lines 196-197: How exactly does a smaller accelerating voltage (12.5 kV) ease the particle morphology problem? Please explain.

Lines 212-213: "...which not only includes particle but also the substrate."

Lines 213-214: Please rephrase "they do only indirectly represent an amount of matter with respect to the particle."

Lines 217-218: "...while the remaining uncertainty originates mainly from the particle to particle variation."

Lines 222-223: why is the case for 20 kV is shown instead of 12.5 kV?

Line 255: "...separately is an important task."

Line 328: here may suffer?

Lines 349-350: "chemically aggressive environments" is too vague. Please elaborate. Representing bio-available iron by spherical surface area and metal oxide mass fraction

alone may be misleading and needs further explanation about its limitations, since Fe dissolution is also highly dependent e.g. on pH and presence of inorganic and organic ligands (which cannot be addressed by the current analytical approach).

Line 659: As a result,

Line 757: e.g.

Line 778: "...while a strict disambiguation can't be done for elements also found in sea salt."

Line 779: "most likely derived only from dust"

Line 790: Western Africa

Line 894: considerably higher

Line 896-908: With regard to the change in dust behavior due to internal mixing, I generally have no objection about the main conclusion that the mixing of dust with sea-salt and sulfate would significantly affect its deposition velocity as stated in the preceding paragraph. I also admit that the conclusion is well supported by the solid results shown in this study. In contrast, the whole idea about enhanced ice-nucleation efficiency of dust through internal mixing needs more careful discussion and is not supported by sufficient results. For example, some studies report deactivation of ice nuclei due to atmospherically relevant coating (e.g. Cziczo et al., Environmental Research letters, 2009), and this deactivation by coating may be more pronounced especially for deposition mode ice nucleation. This impact of coating on different freezing modes might need to be explained clearly in the text not to confuse the readers. Also, the heterogeneous ice freezing temperatures of dust can decrease with increasing concentration of different solutes (e.g. Zobrist et al., J. Phys. Chem. A, 2008) in immersion freezing. There may be a regime where cloud droplets are not dilute enough such that freezing temperatures of droplets activated upon sea-salt / dust mixture can be significantly decreased (Iwata and Matsuki, ACP, 2018). There could be several competing effects

of coating on dust ice nucleation efficiency and this may not act identically for different freezing modes. Besides, as authors pointed out themselves (in line 890), mixed particles with a smaller contribution of hygroscopic material, which remain undetected by the current analytical approach, may be present. Droplet growth and activation kinetics may behave differently at sub- and super-saturated conditions and such seemingly uncoated dust may as well be activated as cloud droplets under higher supersaturation. This subject may need a whole new paper to be discussed. Since there are not much results to support the proposed hypothesis, this whole section about the proposed enhancement of dust ice nucleation efficiency may even be omitted from the current manuscript.

Fig. 18: In this context, isn't the annotation for the third figure from the top supposed to be "sea-salt/silicate mixtures" and "sulfate / dust mixtures"?

Line 1468: "size class is shown in conjunction with. . ."

---

## Author Comment (AC1) · 30 Aug 2018

**General introduction**

We thank the reviewers for their constructive comments.

As both reviewers raised partly the same issues, we will answer the comments below together.

**Major changes to the manuscript are**

a) the restructuring with respect to the long methods section, of which there is now only a summary left in the main manuscript, whereas the major part was moved into the electronic supplement;
b) a necessary modification of the small-iron-particle discussion, which originated from additional later analyses. The latter revealed a previously unknown artifact of the sample substrates. Details are given in the according sections;

c) introduction of a section dealing with feldspar particles in manuscript and supplement.

Our answers are written in red color, and changes to the manuscript are also shown in red.

Below the answers, please find the modified manuscript with figures, and below that we have attached the modified electronic supplement, as this was also discussed by the reviewers.

**Reviewer 1**

**General comments**

The authors have provided a very nice manuscript of measurements and associated modelling of aerosols in the Caribbean boundary layer which is suitable for publishing in ACP. The composition and mixing state of dust, sea salt and sulphate are analysed and they have been detailed in their efforts and provide a good analysis of uncertainty in the results. The general flow of the paper could be improved by condensing the methods and some small additions to the discussions. Of note I feel is the dry deposition discussion – a parameter which hinders aerosol modelling in general and, as shown here, needs further work in constraining.

**Specific comments**

**Methods:**

The methods section is highly detailed, particularly in its use of equations. While it is commendable to be thorough in an analysis I feel the manuscript would greatly benefit from condensing the main methods to the core equations and text. Following each one through in detail to its final derivation is not necessary for the discussion and instead detracts the reader from getting to said results. The SI is the ideal place for much of this text, and some could even be removed totally.

The deposition model section has been condensed to a summary of the most important points and literature references. The original section has been moved to the electronic supplement.

For example, Section 2.3.5: It is good to be thorough and test other statistics, but this is not necessary in the main text and somewhat disturbs the flow. It is sufficient to briefly state you performed the bootstrap analysis with 10,000 replications and move on.

**The statistics section also has been condensed to a summary and the original section has been moved to the electronic supplement.**

Section 2.4.1: The Petters manuscript and hygroscopicity term is well known within the atmospheric community. Small statements about assumed values and uncertainties would be sufficient I feel.

Also changed as requested.

**Results:**

Section 3.2.1: In Figure 5, model d (Aluko) looks like it would perhaps give closer agreement to results in Table 2 than it does. By comparing results of model e (Piskunov) and d (and others if appropriate, e.g. from the tuning exercise) could the Authors also comment on impacts that uncertainty in differences in the deposition velocity for fine (<2.5 um) or coarse (>2.5 um) mode aerosol has on the results? - which are significant for fine mode between models d and e, but more similar in coarse mode.

As the fine mode aerosol only contributes about 2% to the total mass deposition (Fig. 14 of the original manuscript), the uncertainty here would not affect considerably the mass flux. Instead, the mass flux uncertainty is mainly depending on the coarse mode. The situation is slightly different for the mass concentration (Fig. 11 of the original manuscript), as the fine mode of course contributes more here. Nevertheless, also the mass concentration is dominated by larger particles, so the effect is small.

However, due to the model uncertainty, it can't currently be recommended to use deposition measurements for conclusions on atmospheric fine mode or total number concentrations.

Section 3.3.2: This is a great section but would befit from a bit more explanation I feel. How do the airmass providences link to the trajectories? Fig 15 looks similar to the trajectories, but Fig 16 (particularly the iron) has a very difference providence – how was this reached? This is important as the air mass trajectories suggest there is a difference in 2016 in that it has a much stronger European component to it than 2013. Which makes sense with the SO4 sources analysis.

Airmass providence is calculated from the trajectories as outlines in section 2.2, i.e. it reflects frequency of trajectories from a certain area. Also Fig. 16 in the old manuscript was based on the same trajectories. By weighting with different values from the, analysis, other regions can get into focus. That can change the pattern, in particular for rare occurrences. However, ...

However, the southern African source the Authors state for the combustion iron is less obvious I feel from the trajectories themselves, although apparent in Fig 16. Furthermore, the total iron to dust correlation is unfortunately not shown but, as the authors point out, combustion iron has become a topic of much discussion recently and so it would therefore be good to see this result and then put in the context of whether a combustion iron source is visible in 2016 vs 2013. For a south American source can the Authors identify if this likely to be anthropogenic or biomass burning dominated?

... the situation for the iron displayed in Fig. 16 lower part in the old manuscript was different.

As the iron was important to both reviewers, but the pattern was not so easy to explain, we performed additional analyses on the according particles. Also, we tried to make sure that the values were real by using additional blank samples.

At this point it turned out that some of the charges of carbon adhesive suffered from an iron contamination, which we had not observed on these substrates before. The manufacturer and type did not change and not all charges were affected, so it must have been a new source of contamination during the manufacturing or packaging process.

The contamination is on a very low level. If there is a dust event, it becomes completely negligible in comparison to the dust particles (< 1 % of the iron in dust). However, during a few days in 2016 we had very clean conditions with respect to dust, i.e. when the airmasses arrived from Southern America. In these cases, the contamination became visible. As a result, the lower panel of Fig. 16 in the old manuscript showed effectively the source of clean airmasses arriving in Barbados, but not of anthropogenic iron.

Therefore, we have removed the according section from the manuscript and added a comment on artifact removal in the methods section. Also, we have modified the affected plots.

Section 3.3.4: Small additional discussions about 1) Iron solubility: interstitial and cloud-borne changes with sulphate in particular.

We have added a comment on that in the conclusions section

2)Relative concentration of feldspar in the ice nucleation discussion.

We have introduced a 'feldspar index' to determine feldspar content in the methods section and give a plot now of relative feldspar contribution to the silicate fraction, which appears to be relatively constant over time at Barbados.

3)Wet deposition as a loss process when activated to be CDNC. Would be interesting.

We have added a comment on that in the conclusions section, but further work including cloud modeling would be needed to assess these processes.

**Technical corrections**

**Figures**

Please add legends to all figures where missing and check the use of colours is appropriate (see below for some examples).

Fig 2: The use of a log scale and a continuous colour bar is not intuitive. Please change to a discrete colour bar (10 or 5 colours).

**Changed as requested.**

Fig 4: colours do not match numbers using this scale, e.g. green is 2.5-4.5, but blue is 1.2-1.8?

The colors carry additional information, as described in the caption, and do not apply to the x or y scale.

Fig 12: The empty box plot for high volume sampler concentration looks odd. I suggest replacing with a simple horizonal line or a shape (e.g. a star).

We have tried both alternatives, but the plot becomes more difficult to read, as the symbols and lines blend with the existing symbols. Therefore, we would like to keep the plot as it is.

Fig 13: the x-axis is logscale, but the bars are a fixed width. This does not make sense. Either alter to scalar plot (preferred) or alter the width of the bars to match the scale.

The bars perfectly match the scale. They appear as same width because the intervals were chosen equidistantly.

Figs 15, 16: Again, the use of continuous colours would probably be better as discreet.

Changed as requested. Fig. 16 had to be partially removed (see general iron discussion).

Fig 16: 2013 small Fe-rich particles looks like it missing the data?

Values were just extremely low, but Fig. 16 had to be partially removed (see general iron discussion).

Fig S4, S5. Increase legend size to a single bar and add numbers to it.

Changed as requested.

Fig S9 : see Fig 12 note above

See comment above.

Supplementary tables and figures can be grouped together (currently tables are interspersed throughout figures)

Changed as requested.

**Text**

Italics are suggested additions to text.

L35: Largest by mass only, not number.

**Corrected.**

L36-37: Expand this by a sentence, too brief.

Done as requested.

L40: Define the Central American Dust Barrier causes.

Mainly removal, but also meridional transport processes. Added as explanation to the manuscript.

L42: What processes are not fully understood? Use of 'these' is too vague here.

Actually down-mixing as well as deposition processes, but only the latter are topic of this manusrcipt. Clarified.

L44: Change to: '... by physical and chemical processing, ... "

Changed as requested.

L46-47: This is quite obvious. Best to either expand or remove sentence.

Sentence removed.

L47-48: Link with expanded L36-37 as to why this happens.

We have added explanations and examples.

L49: While could start a new paragraph.

Changed as requested.

L49-51: Brief summary of these studies/anything of importance to note?

These studies describe aerosol mixing at different places around the world for certain situations. We have added two sentences to describe the advances of the present manuscript.

L55 and on: change methodical to methods

Changed to methodological by request of the other reviewer.

L62: 'Offline' can mean many things. Define it here to avoid ambiguity.

Parenthesis inserted, defining as 'analysis of aerosol particles collected on a substrate'

L111: remove space before comma

Corrected.

L198: '... mentioned correction methods as a function ....'

Changed as requested.

L199: 'a higher accuracy in . . .. can be achieved'

'in quantification' added

L246: '. . .not the focus ..'

Corrected.

L255: '. . .separately an important. . .'

Corrected.

L260: without seeing the dust: iron ratio it is not possible to say that Fe is safely assumed to be from dust and not combustion.

The term 'dust' in this context of calculating single particle composition serves to differentiate between sea-salt, sulfate compounds and 'dust' inside a single particle. It therefore means an inorganic, thermally stable, non-carbonaceous compound. We do not claim that it is 'natural mineral dust'. In this context, in fact we would also refer to stable anthropogenic iron compounds as 'dust'. We have inserted a parenthesis with the above-mentioned definition.

As additional information: when we look at the single particle scale (after removal of the contamination artifacts), 95 % of the particles have a Fe:Si ratio < 0.33. Only 1.8 % have a Fe:Si ratio > 1, and of these most contain also Si. Therefore we would not assume them to be of anthropogenic origin.

L540: 'conclude'

Corrected.

L722-723. Deposition velocities are described in section 2.4.2.

**Links corrected.**

L792-793: 2016 listed twice. However, I can see no obvious difference in the air masses coming from South America in Fig.S6 anyhow?

**Sentence removed, as indeed not significant.**

L801-803: Nitrate as well as sulphate associated with dust sources is likely to be from Europe (e.g., anthropogenic in origin).

Changed as requested

**Reviewer 2**

**General comments:**

This manuscript presents new methods for accurately determining composition and mixing states of individual coarse aerosol particles by an automated SEM-EDX, and their application on the aged Saharan dust samples collected in the Caribbean boundary layer. Traditionally, individual particle analysis suffered from uncertainties related to poor counting statistics. There has been a significant gap between the ambient mass size distribution and such qualitative information based on individual particle analysis.

The authors however, succeeded to analyze large number of particles by the use of a novel automated SEM-EDX, and to provide a much more comprehensive and quantitative view on the mixing states of dust as a function of particle size. Sampling and data analysis were done very carefully by taking into account many potential sources of errors and uncertainties (e.g. quantification of elements, estimation of dust mass, sampling artifacts, counting statistics) and they are well defined and characterized in the manuscript. I therefore believe the manuscript deserves certain credit and can be a good contribution to ACP. However, such thorough and detailed verification of the methods in turn made the manuscript rather lengthy (especially the method section) and not easy for the readers to digest its major findings. It requires an extensive restructuring, and there are also some concerns on the assumptions made and interpretation of the results. The paper may be recommended for publication after these concerns are properly addressed.

We have moved considerable parts into the electronic supplement and provide a summary in the main manuscript.

**Specific comments:**

Lines 17-19: Please rephrase "Techniques were developed to conclude from collected aerosol on atmospheric concentrations and aerosol mixing state, and different models were compared."

Rephrased to "Techniques are presented and evaluated, which allow for statements on atmospheric aerosol concentrations and aerosol mixing-state based on collected samples."

Line 51: "Data basis is still limited" can be elaborated to make this study stand out more from previous studies.

We have added two sentences on the major differences (methodological advancements, much higher particles numbers (statistical significance) and longer observed time periods).

Line 56: methodological?

Changes as requested (in multiple places)

Lines 68-623: While introduction section is made up of only 3 paragraphs, method section accounts for the majority of the manuscript and the overall structure appears rather unbalanced. Many details, equations and figures in the method section can be moved to supplementary information (SI). The explanation in the text can be significantly shortened by simply referring to corresponding sections in the SI.

We have moved considerable parts into the electronic supplement and provide a summary in the main manuscript.

Line 69: Sampling time and duration for three different samplers are not well explained. How do they coincide? I imagine deposition sampler requires longer time.

We have added remarks on sampling duration to each sub-section, and in addition we provide a table of all sampling times and durations in the electronic supplement (S 1). The sampling time for the DPDS is around 1 day, whereas the active techniques are in the range of 1 hour. In case of more rapidly changing aerosol conditions, this may hamper strict comparability. We have added a remark on this topic.

Also, we have added a comment in section 3.2.1. However, as the same qualitative differences between the techniques – similar in quantitative disagreement – are observed for all samples, it appears to us as a minor source of error.

Line 147: As a result,

**Changed as requested**

Lines 175-178: Validity of estimating volume equivalent diameter (in section 2.3) based only on 2dimensional projected area and perimeter is questionable. It may work rather well for dry and solid particles such as pure silicates, but if particles were internally mixed with soluble materials and sampled at high relative humidity conditions (which might often be the case at Ragged Point), flattening of deliquesced particles on the substrate may become a source of significant sizing bias. Some people employ tilting of sample stages or apply shadowing to measure the height of the particles under SEM analysis.

It is correct that sea-salt particles collected at Ragged Point are nearly always in deliquesced state. However, we have observed frequently crystallized particles instead of the flattened ones mentioned by the reviewer. The flattened ones we did observe for example in the Mediterranean for smaller particles. This might be linked to the size range (large droplets) and to the substrate (hydrophobic), which might favor a compact recrystallization, but also possibly to the composition (potentially low in organics).

In addition, we have checked the average backscatter electron image brightness for different particle classes. We find here that the sea-salt particles show a higher average brightness than the silicate

ones, as predicted by their higher average atomic number (e.g., Goldstein 2003, "Scanning Electron Microscopy and X-ray analysis", chapter 4.4.2). If the particles would be flat, it would be expected that the BSE brightness would be very low, as then the low-Z background would average with a thin film of NaCl to a low brightness.

We have added a note to the according section.

Line 185: achieved instead of reached?

**Changed as requested**

Lines 196-197: How exactly does a smaller accelerating voltage (12.5 kV) ease the particle morphology problem? Please explain.

Fig. 1 was extended to show the increased deviations at higher acceleration voltages.

Lines 212-213: "...which not only includes particle but also the substrate."

**Sentence was modified**

Lines 213-214: Please rephrase "they do only indirectly represent an amount of matter with respect to the particle."

**Sentence was modified**

Lines 217-218: "...while the remaining uncertainty originates mainly from the particle to particle variation."

**Changed as requested**

Lines 222-223: why is the case for 20 kV is shown instead of 12.5 kV?

Two reasons: many other studies use 20 kV as standard, and because problems are more pronounced and better visible at 20 kV than at 12.5 kV (shown above). Fig 1 and according caption and text were modified to demonstrate the dependence on acceleration voltage.

Line 255: "...separately is an important task."

Changed as requested

Line 328: here may suffer?

Changed as requested

Lines 349-350: "chemically aggressive environments" is too vague. Please elaborate.

We have added an example for acidic low pH, but think that the details are well-described in the referenced works and not topic of the present manuscript.

Representing bio-available iron by spherical surface area and metal oxide mass fraction alone may be misleading and needs further explanation about its limitations, since Fe dissolution is also highly dependent e.g. on pH and presence of inorganic and organic ligands (which cannot be addressed by the current analytical approach).

We have added a note to emphasize this point.

Line 659: As a result,

Changed as requested

Line 757: e.g.

Changed as requested

Line 778: "...while a strict disambiguation can't be done for elements also found in sea salt."

Changed as requested

**Line 779: "most likely derived only from dust"**

Changed as requested

Line 790: Western Africa

Changed as requested

Line 894: considerably higher

**Changed as requested**

Line 896-908: With regard to the change in dust behavior due to internal mixing, I generally have no objection about the main conclusion that the mixing of dust with sea-salt and sulfate would significantly affect its deposition velocity as stated in the preceding paragraph. I also admit that the conclusion is well supported by the solid results shown in this study. In contrast, the whole idea about enhanced ice-nucleation efficiency of dust through internal mixing needs more careful discussion and is not supported by sufficient results. For example, some studies report deactivation of ice nuclei due to atmospherically relevant coating (e.g. Cziczo et al., Environmental Research letters, 2009), and this deactivation by coating may be more pronounced especially for deposition mode ice nucleation. This impact of coating on different freezing modes might need to be explained clearly in the text not to confuse the readers. Also, the heterogeneous ice freezing temperatures of dust can decrease with increasing concentration of different solutes (e.g. Zobrist et al., J. Phys. Chem. A, 2008) in immersion freezing. There may be a regime where cloud droplets are not dilute enough such that freezing temperatures of droplets activated upon sea-salt / dust mixture can be significantly decreased (Iwata and Matsuki, ACP, 2018). There could be several competing effects of coating on dust ice nucleation efficiency and this may not act identically for different freezing modes. Besides, as authors pointed out themselves (in line 890), mixed particles with a smaller contribution of hygroscopic material, which remain undetected by the current analytical approach, may be present. Droplet growth and activation kinetics may behave differently at sub- and super-saturated conditions and such seemingly uncoated dust may as well be activated as cloud droplets under higher supersaturation. This subject may need a whole new paper to be discussed. Since there are not much results to support the proposed hypothesis, this whole section about the proposed enhancement of dust ice nucleation efficiency may even be omitted from the current manuscript.

We agree with the reviewer that the topic of mixed particle ice activation will need considerable additional investigations, as we wrote in the conclusions. With the hypothesis, our goal was to point out another process which might be of atmospheric relevance, and not to describe the process in detail (which would be far beyond the scope of the current paper).

However, as we can't provide evidence for a particular process, we have removed the according paragraph from the result section. Instead, we have added a short comment on potential implications at the end, calling for further investigations.

Fig. 18: In this context, isn't the annotation for the third figure from the top supposed to be "sea-salt/silicate mixtures" and "sulfate / dust mixtures"?

Corrected, this was a mistake.

Line 1468: "size class is shown in conjunction with. . ."

Changed as requested

**Composition and mixing state of atmospheric aerosols determined by electron microscopy: method development and application to aged Saharan dust deposition in the Caribbean boundary layer**

Konrad Kandler1,\*, Kilian Schneiders1, Martin Ebert1, Markus Hartmann1,+, Stephan Weinbruch1, Maria Prass2, Christopher Pöhlker2

1Institute for Applied Geosciences, Technical University Darmstadt, 64287 Darmstadt, Germany 2Max Planck Institute for Chemistry, Multiphase Chemistry Department, 55128 Mainz, Germany +now at: Experimental Aerosol and Cloud Microphysics Department, Tropos Leibniz-Institute für Tropospheric Research (TROPOS), 04318 Leipzig, Germany

\*Correspondence to: K. Kandler (kandler@geo.tu-darmstadt.de)

Abstract. The microphysical properties, composition and mixing state of mineral dust, sea-salt and secondary compounds were measured by active and passive aerosol sampling followed by electron microscopy and X-ray fluorescence in the Caribbean marine boundary layer. Measurements were carried out at Ragged Point, Barbados during June/July 2013 and August 2016. Techniques are presented and evaluated, which allow for statements on atmospheric aerosol concentrations and aerosol mixing-state based on collected samples. It became obvious that in the diameter range with the highest dust deposition the deposition velocity models disagree by more than two orders of magnitude. Aerosol at Ragged Point was dominated by dust, sea-salt and soluble sulfates in varying proportions. Contribution of sea-salt was dependent on local wind speed. Sulfate concentrations were linked to long-range transport from Africa / Europe and South America / Southern Atlantic Ocean. Dust sources were located in Western Africa. The dust silicate composition was not significantly varying. 3 % of the silicate particles were pure feldspar grains, of which about a third were K-feldspar. The total dust deposition observed was 10 mg m-2 d-1 (range 0.5–47 mg m-2 d-1), of which 0.67 mg m-2 d-1 was iron and 0.001 mg m-2 d-1 phosphorus. Iron deposition was mainly driven by silicate particles from Africa. Dust particles were mixed internally to a minor fraction (10%), mostly with sea-salt and less frequently with sulfate. It was estimated that average dust deposition velocity under ambient conditions is increased by the internal mixture by 30-140 % for particles between 1 and 10 µm dust aerodynamic diameter, with approximately 35 % at the mass median diameter of deposition (7.0 µm). For this size, an effective deposition velocity of 6.4 mm/s (geometric standard deviation of 3.1 over all individual particles) was observed.

**1** Introduction**

Mineral dust and sea-salt are globally the most abundant aerosol types by mass in the atmosphere (Andreae 1995; Grini et al. 2005). They are considerably affecting the earth's radiation budged (Liao et al. 1998; Choobari et al. 2014) by scattering and absorbing solar and terrestrial radiation. Moreover, they modify cloud processes by supplying condensation nuclei and changing the atmospheric stability conditions (Koehler et al. 2009; Tang et al. 2016; Karydis et al. 2017). Over the North Atlantic Ocean, large amounts of dust are transported westwards in the Saharan Air Layer, until they reach the Caribbean (Karyampudi et al. 1999; Prospero et al. 2014). Here, dust usually does not cross the 'Central American Dust Barrier' to the west. Instead, is mainly removed from the atmosphere, but to a lesser extent also transported in meridional directions (Nowottnick et al. 2011). With respect to the removal, dust becomes mixed down into the marine boundary layer by turbulent and convective processes. Here, it is then subject to wet and dry deposition processes, which remove it from the atmosphere. These deposition processes, however, are not yet fully understood (Prospero et al. 2009; Nowottnick et al. 2011).

During its transport, mineral dust may undergo modifications by physical and chemical processing, cloud processing or microphysical effects (Andreae et al. 1986; Falkovich et al. 2001; Matsuki et al. 2005; Sullivan et al. 2007a; Sullivan et al. 2007b). These processes will change the composition and particle size of dust, and thus modify its radiative properties and cloud impacts. For example, an addition of a soluble compound to an insoluble dust particle may obviously on one hand alter its cloud droplet activation properties (Wurzler et al. 2000; Garimella et al. 2014). On the other hand, it might de-active the original dust particle for deposition ice nucleation (Cziczo et al. 2009). In addition, the coating may on one side enhance the scattering of the particle in dry state by adding non-absorbing material and increasing its size (Bauer et al. 2007; Li et al. 2009), but on the other side in deliquesced state the water shell may increase absorption (Bond et al. 2006), enhancing the absorption of according dust components (Lack et al. 2009).

To assess the mixing state of mineral dust, techniques considering single particles are required. While there have been investigations in the past (Zhang et al. 1999; Zhang et al. 2004; Dall'Osto et al. 2010; Deboudt et al. 2010; Kandler et al. 2011a; Fitzgerald et al. 2015), the data basis is still limited. In particular previous studies based on electron microscopy did not take into account methodological problems. Also, they observed smaller particle numbers, affecting statistical significance, and used shorter observation periods. Studies based on single-particle mass spectrometry, in contrast, were not able to quantify elemental contributions to single particles and therefore could not conclude on material fluxes. In the present work, we present results from two field campaigns in summers 2013 and 2016, where the aerosol in the marine boundary layer at Ragged Point in Barbados was collected by active and passive sampling techniques.

[revised manuscript text omitted]

$$SIAI = \frac{Fe_{oxide}}{m_{dust}} \ 4\pi d_{\nu,dust}^2 \tag{6}$$

with  $Fe_{oxide}$  $m_{dust}$  $d_{v,dust}$  iron oxide mass estimated as  $Fe_2O_3$ , the dust elements oxide mass (refer to S.2 in the electronic supplement) the volume-equivalent diameter of the particle dust fraction.

It should be noted that this approach is of geometrical nature only and does not take into account environmental factors like pH and presence of ligands.

**2.3.5 Statistical uncertainty of total volumes / masses and relative number abundances from single particle measurements**

When assessing the uncertainty of values based on counted occurrences, frequently the counting statistics are assumed to follow a Poisson distribution. However, when calculating total aerosol masses or volumes, besides the measurement errors in particular the – usually few – large particles can introduce a considerable statistical uncertainty, which is not necessarily accounted for by the distribution assumption. Therefore, estimates of the statistical uncertainty based on single particle counts for an a priori unknown frequency distribution (i. e. the counting frequency distribution modified by the also unknown particle size distribution) either require reasonable assumptions or distribution-independent estimators. In the present work, the uncertainty is estimated by a bootstrap approach with Monte Carlo approximation (Efron 1979). For the bootstrap approach, a considerable number of data replications are necessary (Carpenter et al. 2000; Pattengale et al. 2010). On the actual number, different recommendations exist with more than 1000 being among

the most common (Carpenter et al. 2000). As higher numbers lead to smaller errors in the uncertainty estimate, 10,000 replications for each sample were performed in the present work. A comparison of the results of the generally robust bootstrap approach (Efron 2003) to a more simple approach, where the counting statistics is assumed to follow a Poisson distribution, is given in section S.3 of the electronic supplement.

For the assessment of the confidence interval of relative counting abundances, a confidence interval based on a binomial distribution is used as estimate (Agresti et al. 1998), i.e. for a relative number abundance of a certain particle type class r the two-sided 95 % confidence interval is approximated (Hartung et al. 2005).

**2.4 Collection efficiency and deposition velocity relating atmospheric concentrations to deposition rates**

**2.4.1 Determining the size distributions from the free-wing impactor measurements**

Obtaining the atmospheric size distribution and representative contributions of particle populations with different hygroscopicity from the FWI requires two corrections. These corrections are applied to each single particle as a function of its size and composition and the thermodynamic and humidity conditions during sampling. First, a window correction is applied, accounting for the exclusion of particles at the analysis image border (Kandler et al. 2009). Second, the FWI collection efficiency is corrected. For the detailed formalism, refer to section S.4 in the electronic supplement.

[revised manuscript text omitted]
. Higher area coverage, as to be expected, yields higher mixture probability. In particular, if components are present in equal abundances, mixing probabilities become high already for covered area fraction of a few percent. As an example, Fig. S 7 in the electronic supplement shows the upper 95 % confidence limit – i.e. the detection limit for mixtures – of apparent fractions of internally mixed particles for a two-component system as function of source component ratio and area coverage for detectable strong internal mixtures (refer to section 2.5; data are given also in the electronic supplement, Table S 4 and Table S 5). No significant mixture for submicron particles occurs in these cases. Note also the different size maximum for strong versus detectable mixture.

[revised manuscript text omitted]

Recently, the impact of mineral dust composition on clouds via the ice phase has attracted attention. Feldspar particles and in particular K-feldspars are discussed as most efficient ice nuclei (Atkinson et al. 2013; Augustin-Bauditz et al. 2014; Harrison et al. 2016). Therefore, Fig. 13 shows the total feldspar and K-feldspar number fractions with respect to all silicates as determined by the feldspar indices. In general, approximately 3 % of the silicates are pure feldspar particles and slightly less than 1 % K-feldspars. No significant variation is visible for the different periods and years at Barbados, whereas particles collected in Morocco (Kandler et al. 2009) showed slightly higher values. In this respect, the dust composition at Barbados is constant over time. Note that the bulk feldspar contents of the samples might be higher, as the applied technique only detects pure feldspar grains.

**3.3.2 Airmass history and potential aerosol sources**

The airmass provenance of the sampling periods in 2013 and 2016 is generally similar. The trajectories mostly followed the trade-wind path from North-West Africa and East Atlantic Ocean to Barbados (Fig. S 8 in electronic supplement). In 2013, the air was coming more frequently from Western Africa than in 2016. After the tropical storm Chantal in 2013, the airmass origin shifted slightly to more southern regions. In a few cases in 2013, air from the North-West Atlantic Ocean was recirculated into the trade-wind path.

The sea-salt deposition rates are not linked to air mass provenance (not shown). The dust provenance for both years (Fig. 14 a, b) is – as expected – pointing to West Africa. This source region is also identified by isotope measurements in July/August 2013 (Bozlaker et al. 2018). The soluble sulfate deposition (Fig. 14 e, f) is generally linked to three regions, the Atlantic Ocean, West Africa and south west Europe. In particular in 2016, the sulfate sources appear to be located more in Europe and less in Africa. The relative ion balance (Fig. 14 g, h) shows mostly slightly negative values indicating presence of  $NH_4^+$  or  $H^+$ . Interestingly, a positive ion excess is observed for European sulfate in 2016, indicating possible presence of  $NO_3^-$ . These observations support the hypothesis that nitrate as well as sulfate associated with dust sources is likely to be from Europe (e.g., anthropogenic origin; 
[revised manuscript text omitted]
. The internal mixing likely also increases its efficiency to be activated into cloud droplets (Kelly et al. 2007; Kumar et al. 2011). As a result internally mixed dust particles may be subject to preferential removal by wet deposition. However, more systematic investigations are needed to better understand the mixing processes.
- Also the intensity of chemical processing might be affected by the internal mixing, when the particles are activated more efficiently into cloud droplets. For example, the iron solubility for these particles might increase (Shi et al. 2009).

With respect to a potential cloud impact, the observed fraction of dust mixed with soluble species can be used as input parameters for cloud condensation nuclei parameterizations. Regarding the impact of mixing on dust ice nucleation activity, on one side studies show a deactivation of dust for high solute concentrations (Zobrist et al. 2008; Iwata et al. 2018). On the other side, the more efficient activation into cloud droplets might increase the overall availability of dust for immersion freezing. Further studies are needed to assess and constrain the effects.

**5 Data availability**

[revised manuscript text omitted]

Fig. 18: Effective deposition velocity for all dust-containing particles observed at Ragged Point. The blue curves take into account internal mixing and hygroscopic growth at ambient conditions, whereas the orange only regards the dry dust fraction of the particles. In addition, cumulative mass distribution is shown on the inverted right axis. Particle size is given as aerodynamic diameter for the dust fraction of a particle. For the ambient deposition velocity, the geometric mean for each size class is shown in conjunction with the 1 geometric standard deviation range.

Fig. 01